



# Heterogeneous uptake of ammonia and dimethylamine into sulfuric and oxalic acid particles

Meike Sauerwein[1] and Chak Keung Chan[1,2,3]

[1]Division of Environment, Hong Kong University of Science and Technology, Clear Water Bay, Kowloon, Hong Kong
[2]Department of Chemical and Biomolecular Engineering, Hong Kong University of Science and Technology, Clear Water Bay, Kowloon, Hong Kong
[3]School of Energy and Environment, City University of Hong Kong, Kowloon, Hong Kong

*Correspondence to*: Chak Keung Chan (Chak.K.Chan@cityu.edu.hk)

**Abstract.** Heterogeneous uptake is one of the major mechanisms governing the amounts of short-chain alkyl-amines and ammonia ($NH_3$) gases resident in atmospheric particles. Molar ratios of aminium to ammonium ions detected in ambient aerosols often exceed typical gas phase ratios. The present study investigated the simultaneous uptake of dimethylamine (DMA) and $NH_3$ into sulfuric and oxalic acid particles at gaseous $DMA/NH_3$ molar ratios of 0.1 and 0.5 at 10%, 50%, and 70% relative humidity (RH). Single gas uptake and co-uptake were conducted under identical conditions and compared.

Results showed that the particulate dimethyl-aminium/ammonium molar ratios ($DMAH/NH_4$) changed substantially during the uptake process, which was predominantly influenced by the extent of neutralization and the particle phase state. DMA uptake and $NH_3$ uptake into concentrated $H_2SO_4$ droplets were initially similarly efficient, yielding $DMAH/NH_4$ that were similar to $DMA/NH_3$ ratios. As the co-uptake continued the $DMAH/NH_4$ gradually dropped due to a preferential uptake of $NH_3$ into still acidic droplets. Once the droplets were neutralized, the stronger base DMA displaced some of the ammonium absorbed earlier, leading to $DMAH/NH_4$ that were up to four times higher than the corresponding gas phase ratios. At 10% RH, crystallization of partially neutralized sulfate particles prevented further DMA uptake, while $NH_3$ uptake continued, and displaced $DMAH^+$ after the solid particles were completely neutralized, forming almost pure ammonium sulfate. Displacement of $DMAH^+$ by $NH_3$ has also been observed in neutralized, solid oxalate particles. The results illustrate why in ambient liquid aerosols the $DMAH/NH_4$ can be larger than $DMA/NH_3$, despite of an excess of $NH_3$ in the gas phase; the uptake of DMA to aerosols consisting of crystalline ammonium salts, however is unlikely, even if the gas concentrations of DMA and $NH_3$ are of the same magnitude.

## 1. Introduction

$NH_3$ and short-chain alkyl-amines ($R_3N$) gases are frequently detected in the atmosphere. Total emissions of $NH_3$ largely dominate those of $R_3N$ (Schade and Crutzen, 1995). The characteristic ambient mixing ratios of $NH_3$ and $R_3N$ are in the order of several parts per billion by volume and parts per trillion, respectively (Ge et al., 2011; You et al., 2014; Zheng et al., 2015). Many of the emission sources of $R_3N$ such as agricultural and industrial activities also release $NH_3$ (Behera et al., 2013). Hence, elevated $R_3N$ gas concentrations are likely accompanied by enhanced concentrations of $NH_3$ (Schade and Crutzen, 1995; Zheng et al., 2015). Despite the 2-3 orders of magnitude difference in gas phase concentration, particle phase aminium-to-ammonium ($R_3NH/NH_4$) molar ratios of up to 0.2 have been detected. For instance, average $R_3NH/NH_4$ molar ratios of 0.0045–0.17 were measured in $PM_{1.8}$ in urban and rural continental air masses over urban and rural sites in Ontario, Canada (VandenBoer et al., 2011), 0.02 in $PM_{1.0}$ in an urban area of Arizona, USA (Youn et al., 2015) and 0.23 in particles with a vacuum aerodynamic diameter of 50-800 nm in California, USA (Sorooshian et al., 2008).

Large $R_3NH/NH_4$ ratios in particles below 20 nm have been correlated to enhanced particle formation (VandenBoer et al., 2011; Youn et al., 2015). Laboratory studies (Almeida et al., 2013; Jen et al., 2014), field measurements (Mäkelä et al.,




2001; Kulmala et al., 2013), as well as computational methods (Kurtén et al., 2008; Olenius et al., 2013) have indicated that clusters of sulfuric acid ($H_2SO_4$) and DMA are more stable than clusters formed from $NH_3$-$H_2SO_4$ nucleation, and thus $R_3N$ may contribute more to new particle formation than $NH_3$. Furthermore, $R_3N$ are able to form sulfuric acid-amine clusters by replacing $NH_3$ in ammonium-sulfuric acid clusters (Bzdek et al., 2010a; Bzdek et al., 2010b) and ammonium nitrate

nanoparticles (Lloyd et al., 2009), despite the presence of $NH_3$ in the gas phase (Lloyd et al., 2009). If not directly participating in the nucleation of particles, $R_3N$ can also efficiently partition into clusters and small particles (Kürten et al., 2016) to promote particle growth.

However, $R_3N$ are not only detected in the nucleation mode (Mäkelä et al., 2001; Smith et al., 2008), but also in aerosols

exceeding 100 nm in diameter. In fact, mass loadings of alkylaminium ions ($R_3NH^+$) are the highest in marine particles as well as urban and rural aerosols with a diameter of 140-560 nm (Müller et al., 2009; VandenBoer et al., 2011; Youn et al., 2015). These aminium ions were observed to be internally mixed with sulfate, nitrate or organic acids ( Sorooshian et al., 2008; Müller et al., 2009; Pratt et al., 2009; Smith et al., 2010; VandenBoer et al., 2011; Healy et al., 2015; Youn et al., 2015), suggesting that heterogeneous reactions forming aminium salts are an important sink for atmospheric $R_3N$ (You et al.,

2014; Tao et al., 2016). On the other hand, enhanced $NH_4^+$ concentrations in particles of the accumulation mode typically dominate $R_3NH^+$ concentrations, leading to low observed $R_3NH/NH_4$ molar ratios. A second peak of $R_3NH/NH_4$ in the coarse mode has been reported (VandenBoer et al., 2011; Youn et al., 2015), although the causes of these higher ratios in larger particles are not resolved to date.

Chemical characteristics of $R_3N$ and their salts can deviate significantly from those of $NH_3$ and its salts. For instance, due to the electron donor effect of the alkyl groups, the nitrogen atom of $R_3N$ molecules is more nucleophilic towards hydronium ions, making them stronger bases than $NH_3$ (Breitmaier and Jung, 2005). Furthermore, short-chain methyl- and ethyl-aminium nitrates, chlorides, and sulfates possess higher osmotic coefficients than their ammonium counterparts (Bonner, 1981; Macaskill and Bates, 1986; Chu et al., 2015; Sauerwein et al., 2015; Rovelli et al., 2016), which increases the aerosol

hygroscopicity and liquid water content. Besides, secondary and tertiary aminium sulfates remain in the liquid state even at RH below 3% (Chan and Chan, 2013) and they effectively lower the deliquescence RH of the particles when mixed with ammonium sulfate (Qiu and Zhang, 2012). Particles with a large $R_3NH/NH_4$ ratio are consequently less acidic and could absorb more water than ammonium sulfate (($NH_4$)$_2SO_4$), even at low RH.

Particulate $R_3NH^+$ concentrations in ambient aerosols are positively correlated with particle acidity, liquid water content, and RH (Sorooshian et al., 2008; Rehbein et al., 2011; VandenBoer et al., 2011; Youn et al., 2015). Numerous laboratory uptake experiments of $R_3N$ into $H_2SO_4$ (Wang et al., 2010), ammonium nitrate, sulfate and bisulfate have confirmed such correlations (Lloyd et al., 2009; Bzdek et al., 2010b; Qiu et al., 2011; Chan and Chan, 2012, 2013). However, the conditions in those experiments would also promote heterogeneous uptake of $NH_3$ (Swartz et al., 1999). Although there are many

studies on the uptake of either $NH_3$ (Huntzicker et al., 1980; McMurry et al., 1983; Daumer et al., 1992; Swartz et al., 1999; Hanson and Kosciuch, 2003; Liggio et al., 2011) or $R_3N$ (Bzdek et al., 2010a; Bzdek et al., 2010b; Wang et al., 2010; Bzdek et al., 2011; Qiu et al., 2011; Chan and Chan, 2012) into acidic particles, to date none has been reported on the simultaneous uptake of $NH_3$ and $R_3N$.

Barsanti et al. (2009) were the first to model the relative importance of $R_3N$ (using DMA as a model compound) and $NH_3$ in gas-particle partitioning. They showed that even when $NH_3$ gas phase concentrations exceeded those of $R_3N$ by an order of magnitude, aminium $R_3NH^+$ can still dominate $NH_4^+$ in the aqueous acetic acid particle, due to their stronger basicity. Yet,





owing to the lack of chemical and physical parameters available, the study relied strongly on estimations of group contributions.

The present study is the first to investigate the simultaneous uptake of $R_3N$ and $NH_3$ by acidic particles with analysis of
particle phase composition. We explored the temporal changes in $R_3NH/NH_4$ molar ratios during the uptake of DMA and $NH_3$ into $H_2SO_4$ and oxalic acid ($H_2C_2O_4$) particles at different $DMA/NH_3$ gas ratios (0.1 and 0.5) and RH (10% and 50% RH). We used supermicron particles because they enabled a longer observation window and sufficient mass concentrations for studying the temporal changes in particle composition during the uptake until equilibrium was reached. The results also reveal the influence of the extent of neutralization and the change in phase state on the uptake behavior of both gases. DMA
was chosen as the model $R_3N$ due to its atmospheric abundance (Müller et al., 2009; Rehbein et al., 2011; Hu et al., 2015; Youn et al., 2015) and unique characteristics, such as forming a non-crystallizing DMAS droplet even at low RH (Chan and Chan, 2013) and its higher hygroscopicity than that of $(NH_4)_2SO_4$.

## 2. Methods

Supermicron particles deposited on a hydrophobic substrate were placed in a temperature- and humidity-controlled flow-cell
coupled to a Raman microscope setup (Yeung et al., 2009). DMA and $NH_3$ at low ppm levels were generated by directing a humidified $N_2$ carrier flow through permeation tubes holders containing tubes filled with pure liquefied $NH_3$ and DMA at controlled temperatures. The two gases were mixed and introduced to four cells in parallel. Post reaction samples were analyzed by ion chromatography (IC). $R_3N$ and $NH_3$ gas molecules and aminium ions in solution tend to adsorb on surfaces (Dawson et al., 2014; Hansen et al., 2013; Robacker and Bartelt, 1996). To ensure accuracy of the gas ratio, measures were
taken including conditioning the setup for a prolonged period, separating RH conditioning cells and reaction cells. A schematic of the experimental setup is shown in the supplemental information Fig. S1.

### 2.1 Generation and detection of $NH_3$ and DMA gases

A humidified $N_2$ carrier flow was directed into two electronic mass flow controllers (MKS Instruments Inc., GE50A) at $1000 \pm 10$ cm$^3$ min$^{-1}$ and subsequently introduced into glass permeation tube holders (Fig. S1). The tube holders consisted of
a water coated coil and a chamber containing the permeation tubes of either $NH_3$ or DMA (VICI Metronics, Dynacal), as well as a thermocouple to regulate the temperature to $293.3\pm0.2$ K. Permeation rates were determined gravimetrically to calculate the mixing ratio of each gas.

Combination of the DMA flow (0.15 or 0.9-1.0 ppm) and the $NH_3$ flow (1.8-1.9 ppm) resulted in $DMA/NH_3$ ratios of
$0.07\pm0.01$ and $0.46\pm0.04$ at 10% RH, as well as $0.07\pm0.01$ and $0.49\pm0.02$ at 50% RH (Table 2), and $0.49\pm0.02$ at 10% and $0.52\pm0.01$ at 70% RH for experiments with $H_2C_2O_4$. We denote these experimental conditions for uptake into $H_2SO_4$ by $0.1_{10\%}$, $0.5_{10\%}$, $0.1_{50\%}$, and $0.5_{50\%}$, and for uptake into $H_2C_2O_4$ by $ox0.5_{10\%}$ and $ox0.5_{70\%}$. For co-uptake gas flows from both permeation devices were mixed and split equally into four custom-made PTFE flow cells. Single gas flows were generated by bypassing one of the permeation tubes. The cells were maintained at $296.3\pm1.0$ K and RH of $10\%\pm2\%$ and $50\%\pm3\%$. The
system including the PTFE cells was equilibrated with the $NH_3$ and/or DMA gas for > 12 h before the start of each experiment. The stability of the generated gas concentrations arriving at the PTFE cells was confirmed by ion-molecule reaction mass spectrometry (IMR-MS, AirSense, V&F Analyse-und Messtechnik GmbH).





### 2.2 Particle generation

The stock solutions of 30 wt% $H_2SO_4$ or saturated $H_2C_2O_4$ were prepared from ultrapure water (18.2 MΩ) and concentrated $H_2SO_4$ (97 wt%, Acros Organics, titrated against standardized NaOH) or $H_2C_2O_4$ powder (99%, Aldrich). For each experiment, a few mL of the solution were drawn into a piezoelectric particle generator (MicroFab Tech., Inc.).

Approximately 2000±100 droplets of 60 μm in diameter were deposited on a hydrophobic film (FEP membranes, YSI Inc.) The sample was then inserted into a clean flow cell connected to humidified high purity $N_2$ (~1000 $cm^3min^{-1}$) to equilibrate to either 10 or 50% RH for 45 min. The sample films were subsequently transferred into the PTFE reaction cells. RH conditioning and transfer of films took place inside a glove bag (GLOVEBAG, Glas Col®) to avoid exposure of the samples to room air and humidity.

### 2.3 Particle analysis

For each condition, the experiment was repeated in different time intervals to complete one time series. The samples were removed from the cell and submerged in ~12 mL of ultrapure water for extraction and subsequent chemical analysis. Both cation and anion contents were measured by IC (Metrohm, 881 compact IC Pro) as described by Sauerwein et al. (2015).

IC yields total values for DMA and $NH_3$ species, and hence the distribution between molecules and ions in the samples could not be determined. In the following we use DMAH to represent $nNH_2(CH_3)_2^+ + nNH(CH_3)_{2(aq)}$ and $NH_4$ to represent $nNH_4^+ + nNH_{3(aq)}$ in the particles, where $n$ denotes the molar amounts of each compound. The same is true for the distribution between acidic species. Hereafter the molar amount of the total sulfate is indicated by $SO_4 =$

$nH_2SO_4 + nHSO_4^- + nSO_4^{2-}$, and the total oxalate as $C_2O_4$. Molar ratios in the particle phase are accordingly denoted by $DMAH/NH_4$, $DMAH/SO_4$, and $NH_4/SO_4$, while gaseous molar ratios are denoted by $DMA/NH_3$.

DMAH concentrations of the first measurement points for particles in the $0.1_{10\%}$ and $0.1_{50\%}$ conditions lay at the lower end of the IC calibration range. A conservative estimation of up to 15% uncertainty for these values would decrease the

$DMAH/NH_4$ ratio by 0.02, which has negligible impacts on the observed trends and values (see Fig.3).

After 35-38 hours, samples showed no significant changes in particle composition and were thus considered to be in equilibrium. Equilibrated samples were divided into two groups, with one directly undergoing IC analysis and the other placed in clean cells under an $N_2$ atmosphere for > 5 hours prior to IC analysis to further investigate their compositional

stability.

Furthermore, uptake experiments at $0.5_{10\%}$ and of $NH_3$ at 10% RH were repeated with particles of 60-200 μm in size in flow cells equipped with a quartz window to track alterations in chemical composition and concurrent morphological changes with a Raman microscope (Renishaw RM series) as described by Yeung et al. (2009) and Chu et al. (2015). A 20 mW Argon

ion laser (514.5 nm) was deployed for sample excitation and a 1800 g mm$^{-1}$ grating was selected to obtain spectra in the range of 200 to 4000 cm$^{-1}$ with a resolution of 1.4 cm$^{-1}$.

### 3. Results and discussion

Uptake of $NH_3$, DMA and their mixtures into $H_2SO_4$ particles were conducted at 10% and 50% RH. Single component uptake served as the base case for comparison. Additional experiments with $H_2C_2O_4$ at 10 and 70% RH were conducted at a

gas ratio of 0.5 only and are discussed in Sect. 3. A summary of all the experimental conditions is given in Table 1.





The uptake of alkaline gases into acidic droplets involves a series of inter-related processes including gas phase diffusion, immediate reaction of gas molecules colliding with the aerosol surface (Eq. 1a and b) or adsorption and dissolution (Eq. 2a and b), followed by further liquid phase diffusion and proton transfer in the bulk particle (Eq. 3a and b) (Swartz et al., 1999;

Kulmala and Wagner, 2001; Davidovits et al., 2006; Pöschl et al., 2007; Kolb et al., 2010).

$$NH_{3(g)} + H_3O^+_{(surface)} \rightleftharpoons NH_4^+_{(sf)} + H_2O \tag{1a}$$

$$NH(CH_3)_{2(g)} + H_3O^+_{(surface)} \rightleftharpoons NH(CH_3)_2H^+_{(sf)} + H_2O \tag{1b}$$

$$NH_{3(g)} + H_2O \rightleftharpoons NH_{3(aq)} + H_2O \tag{2a}$$

$$NH(CH_3)_{2(g)} + H_2O \rightleftharpoons NH(CH_3)_{2(aq)} + H_2O \tag{2b}$$

$$NH_{3(aq)} + H_3O^+ \rightleftharpoons NH_4^+_{(aq)} + H_2O \tag{3a}$$

$$NH(CH_3)_{2(aq)} + H_3O^+ \rightleftharpoons NH(CH_3)_2H^+_{(aq)} + H_2O \tag{3b}$$

The above equations and related reaction steps indicate a strong acidity dependence, thus with increasing neutralization, the

scope of DMA and $NH_3$ uptake may change. Here, the extent of (stoichiometric) neutralization of the particles is defined as the number of moles of alkaline species over moles of acidic species, $X = (DMAH+NH_4)/SO_4$ or $X = (DMAH+NH_4)/C_2O_4$ and is hereafter referred to as the neutralization ratio. In the course of the uptake experiments, the neutralization ratio ranged from highly acidic ($X = 0$) to neutral ($X = 2$). But not all equilibrated particles were completely neutralized, as will be further discussed in Sect. 4.

**3.1 Single gas uptake**

At 50% RH, the uptake of $NH_3$ fully neutralized the $H_2SO_4$ droplets within 2 hours, forming aqueous $(NH_4)_2SO_4$ droplets (Fig. 1, grey diamonds). At 10% RH, $NH_3$ uptake (Fig. 1, open diamonds) was similar to that at 50% RH initially, until $X$ exceeded 1.1, where crystallization significantly retarded the subsequent uptake. The continued increase in particulate $NH_4^+$ even after crystallization indicates that crystalline acidic particles were still susceptible to uptake, but imposed bulk diffusion

limitations that retarded the uptake. Neutralization was not complete within the measured period of 15 hours.

Uptake of DMA at the mixing ratio of 0.15 ppm (Fig. 1, triangles) was significantly slower than that at 1 ppm (Fig. 1, squares). Like the initial uptake of $NH_3$, DMA uptake did not differ significantly between 10% and 50% RH, until approaching equilibrium at DMAH/SO$_4$ ratios of 1.5±0.1 at 10% RH, and 1.7±0.1 (DMA$_{0.15ppm}$) and 1.9±0.1 (DMA$_{1ppm}$) at

50% RH (Table 1, $X_{equil}$). There was no indication of phase change in these particles even at 10% RH. This is consistent with earlier studies, where secondary and tertiary methyl and ethyl-aminium sulfates were described as hygroscopic, non-crystallizing salts at RH ≤ 10% (Qiu and Zhang, 2012; Chan and Chan, 2013; Chu et al., 2015). Furthermore, these studies showed that upon drying of synthesized DMAS droplets (DMAH/SO$_4$ ratios = 2) with amine-free air, DMA evaporated from the particles, leading to a final DMAH/SO$_4$ ratio of 1.5 at < 3% RH (Chan and Chan, 2013; Chu et al., 2015). In our

experiment, the same equilibrium DMAH/SO$_4$ ratio of 1.5 was established at 10% RH, despite a continuous supply of DMA gas.

Particles exposed to 1 ppm of DMA at 10% RH were more neutralized initially with $X_{max\ neutral} = 1.7±0.1$ (Table 1,) and then equilibrated to lower $X_{equil}$ of 1.5±0.1 (Table 1). Some of the initially absorbed DMA molecules had re-partitioned into the

gas phase despite the presence of DMA gas in the surrounding atmosphere. The phenomenon of re-volatilization is further discussed in Sect. 4.





### 3.2 DMA-NH$_3$ co-uptake

Figure 2 depicts the temporal profiles of DMAH/SO$_4$, NH$_4$/SO$_4$ and $X$ at the different gas ratios and RH. At 10% RH particles solidified during the experiment (Fig. 2a and b, indicated by crosshatched areas) and needed 2 to > 18 hours (for 0.5$_{10\%}$ and 0.1$_{10\%}$, respectively) to completely neutralize SO$_4$. Upon reaching neutralization, NH$_3$ had almost completely

displaced the DMAH absorbed earlier from the solid particles. The phase transition and DMAH displacement at 10% RH will be further discussed in Sect. 2.3 and 2.4.

At 50% RH SO$_4^{2-}$ was completely neutralized (Fig. 2c and d) within 1-2 hours. Neutralization in droplets was followed by a partial displacement of NH$_4$ by DMA, reaching a final equilibrium composition enriched in DMAH (Sect. 2.2). The results

show that the DMAH/NH$_4$ ratios varied substantially during uptake before stable compositions were reached. In the following, we will discuss the dependence of gas uptake on the phase state and neutralization ratio of the particles.

### 3.2.1 Uptake into liquid acidic droplets

Figure 3 displays the changes in the DMAH/NH$_4$ ratios as a function of time for the four co-uptake experiments. For the first measurement point under each condition, the DMAH/NH$_4$ ratio (Table 2, t$_{initial}$) was close to the gas phase DMA/NH$_3$ ratio

(indicated by grey bands in Fig. 3), implying that initially both gases partitioned equally effectively into highly concentrated H$_2$SO$_4$. For instance, in panels a and b the DMAH/NH$_4$ ratios in the acidic droplets were 0.07±0.01 and 0.43±0.04, comparable to gaseous DMA/NH$_3$ ratios of 0.07 and 0.46, respectively.

Swartz et al. (1999) measured the heterogeneous uptake of NH$_3$ into a chain of 70 wt% and 40 wt% H$_2$SO$_4$ droplets

(equilibrated at 10% and 50% RH, respectively), and obtained gas phase diffusion-corrected uptake coefficients ($\gamma_{NH3}$) of 0.8 and 1.0, respectively. The highly effective uptake into concentrated H$_2$SO$_4$ at pH < 0 was attributed to surface reactions, i.e., NH$_3$ molecules reacting with interfacial hydronium ions (H$_3$O$^+$) without prior solvation (Swartz et al., 1999). As H$_2$SO$_4$ droplets in the present study have a solution pH ≤ - 0.9 (Wexler and Clegg, 2002) at both 50% RH (43 wt% H$_2$SO$_4$) and 10% RH (64 wt% H$_2$SO$_4$), protonation without prior dissolution may take place for NH$_3$ in the first few minutes of uptake when

pH is very low.

To date no systematic study on the relevance of surface protonation has been conducted for DMA uptake. However the gas phase basicity of NH$_3$ and its derivatives (such as R$_3$N) have been shown to correlate well with the differential heats of chemisorption on acidic (zeolite) surfaces (Parrillo et al., 1993). Since DMA possesses a slightly higher gas phase basicity

than NH$_3$ (Brauman et al., 1971; Parrillo et al., 1993), DMA gas molecules might, similar to NH$_3$, have a high affinity to interfacial H$_3$O$^+$. Surface protonation on fresh H$_2$SO$_4$ particles may thus be important for the initial uptake of both NH$_3$ and DMA, which could explain why the initial particle phase ratio is equal to the gas phase ratio.

Wang et al. (2010) reported an uptake coefficient ($\gamma_{DMA}$) of about 0.03 for DMA uptake into concentrated H$_2$SO$_4$ of

≥ 62 wt% (≤ 10% *RH*) at 283 K, which is noticeably smaller than the coefficient of close to unity reported for NH$_3$ uptake into H$_2$SO$_4$ of similar acidity (Swartz et al., 1999). In the current study, NH$_3$ uptake into fresh H$_2$SO$_4$ droplets was not overwhelmingly dominant. However, as the uptake continued, the DMA/NH$_4$ ratios dropped by 30-40% for all experimental conditions within the first 1-2 hours (Fig. 3 a-d), which indicates a preferential uptake of NH$_3$. Since the gas concentrations of both NH$_3$ and DMA were constant, it is likely that the decreasing particle acidity and increasing neutralization ratio

caused this change. Swartz et al. (1999) reported a threshold pH ≤ 0 for surface protonation to occur and a drop in $\gamma_{NH3}$ by one order of magnitude when the pH was increased to above zero. Hanson and Kosciuch (2003) observed a similar drop in $\gamma_{NH3}$ during the uptake of NH$_3$ into H$_2$SO$_4$, when the solution approached ammonium bisulfate (NH$_4$HSO$_4$) composition.




Using the *E-AIM* model (Wexler and Clegg, 2002, *www.aim.env.uea.ac.uk/aim*) we estimated that $n\mathrm{H_3O^+}$ decreased by about 40% as the neutralization ratio increased from $X = 0$ ($\mathrm{H_2SO_4}$ droplets) to 0.5. Once the particles were half neutralized ($X = 1$, bisulfate stoichiometry), the solution pH were estimated to exceed zero. Hence if we assume a similar threshold as reported by Swartz et al. (1999), surface protonation (Eq. 1a and b), which can explain the comparable initial uptake of DMA and NH$_3$, may no longer be relevant when particles approached bisulfate composition.

Figure 4 compares the uptake of DMA and NH$_3$ in single and mixed gas experiments. The uptake trends of single gas and co-uptake did not deviate noticeably, confirming that DMA and NH$_3$ uptake took place independent of each other. Despite the increase in pH when neutralization ratios exceeded unity ($X = 1$), there were sufficient amounts of H$_3$O$^+$ to support the uptake of both gases. Hence, the presence of NH$_3$ gas molecules at 14 times higher concentration than DMA had little effect on the uptake of DMA into acidic particles. The same can be said for DMA, which seemed not to have influenced NH$_3$ partitioning into acidic droplets. This finding clearly differentiates the group of hydrophilic R$_3$N from more hydrophobic organics such as hexadecane, hexadecanol (Daumer et al., 1992) or typical atmospheric organic vapor (Liggio et al., 2011), which form an organic film that limits the access of NH$_3$ to the inorganic core.

The independent uptake of DMA and NH$_3$ at 10% RH prevailed until the particles underwent phase change (Fig. 4a and c, dotted lines). Once particles effloresced, a preferential uptake of NH$_3$ was observed, which resulted in a significant drop in DMAH/NH$_4$ (Fig. 3a and b), as discussed in Sect. 2.3. At 50% RH the co-uptake of DMA and NH$_3$ was independent from each other until the particles reached complete neutralization ($X = 2.0$) (Fig. 4b and d, solid lines). Once neutralized, only DMA uptake continued (Fig. 4b) with concurrent displacement of NH$_4^+$ (Fig. 4d) as discussed hereafter.

### 3.2.2 Displacement of NH$_4^+$ from neutralized droplets

When approaching full neutralization, where both gases started to compete for limited available H$_3$O$^+$ ions, the DMAH/NH$_4$ ratios in droplets of the $0.1_{50\%}$ and $0.5_{50\%}$ experiments started to increase (Fig. 3c and d, solid line). In solution DMA (pK$_a$= 10.64; Hall, 1957) is a stronger base than NH$_3$ (pK$_a$ = 9.21), consequently it has a higher affinity for H$_3$O$^+$ than NH$_3$ does. Thus, while the fraction of DMA species gradually increased, some of the NH$_4^+$ dissociated (Eq. 3a, reverse reaction) and NH$_3$ was released back to the gas phase (Eq. 2a, reverse reaction).

Similar displacement of NH$_4^+$ by alkyl-amines has been reported for aqueous particles of ammonium bisulfate, chloride, oxalate and sulfate at 50 and 75% RH (Chan and Chan, 2012, 2013) and nitrate at 20% RH (Lloyd et al., 2009). Similarly, Lloyd et al. (2009) observed the displacement of NH$_4^+$ by trimethylamine (TMA) from water-coated NH$_4$NO$_3$ nanoparticles despite an excess of NH$_3$ gas ($n\mathrm{TMA_{(g)}}/n\mathrm{NH_{3(g)}} = 0.1$), although they did not report the quantity of NH$_4^+$ displaced. Their gas phase conditions were comparable to our experiment at $0.1_{50\%}$. In our experiment DMA was able to displace about 9% of the initially absorbed NH$_4^+$ (Table 2, $\chi_{\mathrm{NH4+}}$). The equilibrium DMAH/NH$_4$ ratio in the $0.1_{50\%}$ condition was 0.18±0.02 (Table 2, $t_{\mathrm{equil}}$), which indicates an enhancement of DMA by a factor of 2-3 in the particle phase compared to the gas phase. Particles under $0.5_{50\%}$ conditions equilibrated at a DMAH/NH$_4$ ratio of 1.77±0.13, hence 3-4 times higher than the gas ratio; by the time a stable particle composition was established, 50% of the initially absorbed NH$_4^+$ was displaced (Table 2). Yet it should be noted that equilibrated particles at $0.5_{50\%}$ possessed a neutralization ratio of only $X = 1.8$±0.1, hence the particle was incompletely neutralized (see Sect. 4) despite the presence of 1.9 ppm NH$_3$ and 0.9 ppm DMA in the surrounding gas phase.

The experimental equilibrium DMA/NH$_3$ and DMAH/NH$_4$ ratios (Table 2, $t_{\mathrm{equil}}$) were compared with calculations using the *E-AIM* Model (Model II, 296 ± 1 K, no solid formation; Wexler and Clegg, 2002). Measured equilibrium DMAH/SO$_4$, NH$_4$/SO$_4$ were used as input parameters. The *E-AIM* predicted DMA/NH$_3$ gas ratios that would equilibrate over particles lay



with DMA/NH$_3$ of 0.14 and 0.01, below the experimental DMA/NH$_3$ gas ratios of 0.49 and 0.07. Considering the entire uptake process, the modelled result would imply a more intensive displacement of NH$_4^+$ by DMA and stronger enrichment of DMA relative to NH$_3$ species in the particle phase compared to the gas phase. Yet, the experimental findings are qualitatively consistent with the *E-AIM* - modeled values and with earlier simulations by Barsanti et al., (2009), who

reported that DMAH/NH$_4$ in submicron acetic acid droplets can be significantly larger than their gas phase ratio, even if gas concentrations of NH$_3$ dominated DMA by one to three order of magnitudes.

Thus, in the current study we provide experimental evidence that DMA preferentially partitions into neutralized liquid sulfate particles over NH$_3$, due to its stronger alkalinity. DMA then partially displaces NH$_4^+$ from neutralized aqueous

particles even when the NH$_3$ gas concentration is one order of magnitude greater than the DMA gas concentration.

### 3.2.3 Phase transition and uptake into solid acidic particles

NH$_3$ (single gas) uptake into H$_2$SO$_4$ at 10% RH decelerated noticeably when the NH$_4$/SO$_4$ ratio exceeded 1.1 (Fig. 5, filled triangles). A comparable retardation occurred in the co-uptake experiments 0.1$_{10\%}$ and 0.5$_{10\%}$, but at NH$_4$/SO$_4$ ratios of about 1.5 (Fig. 5a and b, open triangles).

We used Raman microscopy (Chu et al., 2015) to further investigate the slowdown of the reactions and possible changes in the physical state for the single uptake of NH$_3$ (Fig. 6) and the co-uptake experiment 0.5$_{10\%}$ (Fig. 7). Note that each sample was composed of several hundreds of closely packed droplets deposited on a substrate. The droplets were not expected to reach the same cation-to-sulfate stoichiometry, nor exhibit phase transitions at the exact same time. For the acquisition of in-situ Raman signals, we selected individual particles just before and right after phase transition, and in crystalline state, to

represent the phase change process.

During the single gas uptake of NH$_3$ into H$_2$SO$_4$ droplets at 10% RH, crystallization occurred within the first 60 min. As shown in Fig. 6, during efflorescence of acidic droplets (Fig. 6, spectra 2 and 3) the HSO$_4^-$ characteristic bands at 590 cm$^{-1}$ and 1035 cm$^{-1}$ (Dawson et al., 1986; Lund Myhre et al., 2003) transformed to doublets at 579/609 cm$^{-1}$ and 1013/1043 cm$^{-1}$

of solid particles (Fig. 6, spectra 2 and 3), suggesting the formation of crystalline NH$_4$HSO$_4$ (Dawson et al., 1986; Colberg et al., 2004). As uptake continued, a gradual shift towards (NH$_4$)$_2$SO$_4$ was indicated by an increase in the SO$_4^{2-}$ stretching mode 975 cm$^{-1}$ and a decrease in the HSO$_4^-$ band at 579 cm$^{-1}$ (Fig. 6, spectrum 4). The retarded diffusion of NH$_3$ from the surface to the interior of the crystals is likely to have limited the uptake, explaining why the spectra did not fully resemble (NH$_4$)$_2$SO$_4$ (Fig. 6, spectrum 5) within the measured period.


Efflorescence was also observed for the co-uptake of NH$_3$ and DMA at 10% RH. Under 0.5$_{10\%}$ conditions, most particles experienced the first morphological change after 40–60 min of uptake, forming fairly spherical solids with long lined patterns (Fig. 7, yellow rectangular). Raman spectral analysis of particles that had only just transitioned from liquid to solid phase state (Fig. 7, spectrum 3) showed an emerging SO$_4^{2-}$ band at 984 cm$^{-1}$. Meanwhile the HSO$_4^-$ frequencies near 590 cm$^{-1}$

$^{-1}$ and 1030 cm$^{-1}$ in the droplet (Fig. 7, spectrum 2) shifted towards 597 cm$^{-1}$ and 1043 cm$^{-1}$ in the solid particle (Fig. 7, spectrum 3) but both remained single broad peaks, without signs of scissoring as observed for bisulfate from NH$_3$ single gas uptake (Fig. 6, spectra 3 and 4). The observed features more closely resemble the spectral characteristics of letovicite (NH$_4$)$_3$H(SO$_4$)$_2$ (Colberg et al., 2004) than those of NH$_4$HSO$_4$ or (NH$_4$)$_2$SO$_4$.

The during NH$_3$-DMA co-uptake the absorbed DMAH$^+$ seems to have suppressed NH$_4$HSO$_4$ precipitation in particles with a composition of 1.1 < NH$_4$/SO$_4$ < 1.5, so that the phase change started only at a NH$_4$/SO$_4$ around 1.5 (Fig. 5a and b). It can be seen from the 0.1$_{10\%}$ experiment (Fig. 5a) that this suppression of NH$_4$HSO$_4$ precipitation also occurred if the DMAH/NH$_4$ at





the time of phase change was as low as 0.05 (Table 2, $t_{pc}$), hence even if the concentration of DMA species was low compared to the amount of $NH_3$ species present in the particle. Since $R_3NH/NH_4$ ratios in ambient samples near emission sites can be up to 0.23 (Sorooshian et al. 2008), and with $DMAH^+$ as the most frequently detected aminium ion in aerosols (Müller et al., 2009; Rehbein et al., 2011; Hu et al., 2015; Youn et al., 2015) $DMAH/NH_4$ ratios in aerosols can possibly

reach ≥ 0.05 and thus could change the crystallization behavior of $NH_4$–H–$SO_4$–salts in atmospheric particles.

By retaining the particles of a composition of $1.1 < NH_4/SO_4 < 1.5$ at 10% RH in liquid phase, the presence of DMA species accelerated the uptake of $NH_3$ compared to single gas $NH_3$ uptake where crystallization retarded the uptake into particles with $NH_4/SO_4 > 1.1$.

After the $NH_4$–DMAH–mixed particles crystallized, the uptake of $NH_3$ continued at a slower pace (Figure 5a and b), while DMA was no longer absorbed and the $DMAH/NH_4$ ratio started to decrease (Fig. 3a). In earlier reports of single gas $R_3N$ uptake (where $R_3N$ might be methylamine, DMA, TMA or triethylamine) $R_3N$ (in the absence of $NH_3$) were observed to effectively adsorb onto $NH_4HSO_4$ surfaces in a coated flow reactor (Qiu et al., 2011). In the presence of $NH_3$ in our experiments, however, DMA was not taken up by acidic crystalline particles. Liu et al. (2012) reported steric effects to

influence the uptake effectiveness of primary, secondary and tertiary methyl-amines into solid citric and humic acid measured in a Knudsen cell reactor, where smaller $R_3N$ molecules are more effectively absorbed than larger ones. A similar effect of steric hindrance may have caused the preferential uptake of the smaller $NH_3$ molecules over DMA in our experiments. Besides steric reasons, the release of lattice enthalpy during the formation of $(NH_4)_3H(SO_4)_2$ and $(NH_4)_2SO_4$ may have made the uptake of $NH_3$ also thermodynamically favorable.

DMAH/$NH_4$ ratios at the time of neutralization ($t_{neutral}$) reached $0.03\pm0.00$ for the $0.1_{10\%}$ condition and $0.19\pm0.01$ for the $0.5_{10\%}$ condition (Table 2), showing a slight enrichment of $NH_4^+$ in the particle over the gas phase.

### 3.2.4 Displacement of $DMAH^+$ from solid neutralized particles

After reaching full neutralization, both Raman spectral analysis and IC results of solid particles indicated a gradual reduction

of DMA species and increase of $NH_3$ species in the particle phase, which is also reflected in decreasing $DMAH/NH_4$ ratios in Fig. 3a and b.

10–20 minutes after the phase transition, $DMAH^+$–$NH_4^+$–mixed particles experienced a second morphological change to a polycrystalline structure (Fig. 7, grey rectangular). Raman spectra of these particles showed a slowly vanishing $HSO_4^-$ band

at 1043 $cm^{-1}$. However, a significant decrease in the full width at half maximum (FWHM) of the $SO_4^{2-}$ band at 980 $cm^{-1}$, which indicates the (full) crystallization of sulfate (Lee et al., 2008), was only observed hours after the morphology change (Fig. 7, spectrum 4). We suspect that the observed morphological changes are related to the formation of a $(NH_4)_2SO_4$ shell structure due to $DMAH^+$ displacement by $NH_3$. $DMAH^+$ was subsequently slowly displaced from the particle core, which eventually led to the formation of crystalline $(NH_4)_2SO_4$ with only traces of $DMAH^+$ left inside the particles.

It is interesting to note, that the DMAH/$NH_4$ ratios of equilibrated particles for $0.5_{10\%}$ and $0.1_{10\%}$ were both 0.02 (Table 2). Even if the $NH_3$ gas concentration is only twice of that of DMA, $NH_3$ can almost completely displace $DMAH^+$ from the solid particle.

Since different gas mixtures resulted in similar particle composition, calculations by *E-AIM* based on these equilibrated

particle compositions yielded similar gas phase DMA/$NH_3$ ratios for the $0.1_{10\%}$ and $0.5_{10\%}$ experiments of 0.57 and 0.32, with an uncertainty of up to 40% due to strong temperature sensitivity. The modelled results are qualitatively in agreement




with the experimental results, confirming that despite considerable amounts of DMA in the gas phase, the equilibrated solid sulfate particle would predominantly contain $NH_3$ species.

Overall it can be concluded that, for DMA-$NH_3$ co-uptake, $NH_3$ is favorably absorbed into acidic liquid particles, except for very concentrated $H_2SO_4$, where DMA and $NH_3$ seem to partition similarly effectively. Unless the particles either crystallize or are close to full neutralization, DMA and $NH_3$ in general do not influence each other's uptake and act as if they were the only gas present.

For uptake into neutralized particles, DMA is favorably absorbed into liquid neutral particles due to its stronger basicity leading to partial $NH_4^+$ displacement depending on gas phase $NH_3$ concentration; $NH_3$ is favorably taken up into solid neutral particles and can almost completely displace $DMAH^+$ due to the thermodynamically favorable formation of $(NH_4)_2SO_4$ crystals and steric reasons.

### 3.3 Uptake into oxalic acid particles

We performed additional experiments with $H_2C_2O_4$ to further elucidate the influence of phase state on the co-uptake process. $H_2C_2O_4$ is a model organic acid frequently detected in ambient aerosols (Kawamura and Ikushima, 1993; Decesari et al., 2000; Yao et al., 2003; Yu et al., 2005; Müller et al., 2009) . Efflorescence of aqueous supermicron $H_2C_2O_4$ particles occurs at 51.8%-56.7% RH, forming anhydrous solids (Peng et al., 2001). We chose to examine the co-uptake of DMA and $NH_3$ into initially solid and initially liquid particles, respectively, at 10% and 70% RH.

Figure 8 compares the co-uptake into $H_2C_2O_4$ and $H_2SO_4$ particles at DMA/$NH_3$ = 0.5. Within the first hour of uptake into solid $H_2C_2O_4$ particles at 10% RH (Fig. 8a, inverted triangles), the neutralization ratio increased to $X = 0.3$. In the subsequent 15 hours, no further increase occurred, suggesting that the gas uptake into anhydrous $H_2C_2O_4$ particles was incomplete and may have been limited to adsorption on the surface and outer layers of the particles. $DMAH/NH_4$ ratios decreased from initially 0.26±0.05 during the first four hours of uptake to 0.17±0.04 after 14-15 h (Table 2), despite a relatively constant overall neutralization ratio ($X = 0.3$). Some of the initially ad-/absorbed DMA molecules were displaced by $NH_3$.

The uptake at 70% RH starting with $H_2C_2O_4$ droplets was monitored under the microscope for subsequent phase changes (Fig. S2). Within 15 minutes of uptake, liquid particles transformed into round, irregularly patterned solids, suggesting that the uptake of even small amounts of gas caused a phase change in these $H_2C_2O_4$ particles. The formation of ammonium salts via the uptake of $NH_3$ has earlier been shown to decrease the hygroscopicity of $H_2C_2O_4$ particles and influence their phase state (Peng and Chan, 2001; Ma et al., 2013). We found that prior to phase change, a ring of small satellites formed around the droplets. This observation can be most plausibly explained by a pinning effect caused by fast evaporation of the solvent from a droplet deposited on a substrate (Deegan et al., 1997). Similar halo formation has been described for other atmospheric particles (Hamacher-Barth et al., 2016), yet, we believe that this effect does not have a signifincant impact on the general uptake observations here. The co-uptake trends into solidified $H_2C_2O_4$ particles at the $ox0.5_{70\%}$ conditions were comparable to those of $0.5_{10\%}$ and $0.5_{50\%}$ into $H_2SO_4$ (Fig. 8b), indicating that the solids formed during phase change (dashed lines, Fig. 8) did not retard the uptake and that diffusion inside the particle bulk was not a limiting factor of the uptake. In fact, the particles almost completely neutralized within the first hour of reaction. Similar observations have been reported by Li et al. (2015), who investigated $NH_3$ uptake into submicron organic particles produced from oxidation of isoprene, a precursor gas of $H_2C_2O_4$ (Ervens et al., 2014). Li et al. (2015) showed that even at RH down to <5%, $NH_3$ uptake was not restricted by diffusion. Unlike the co-uptake into partially neutralized $H_2SO_4$ particles at 10% RH, where DMA uptake


discontinued once the particles formed solids, uptake of DMA into solid, partially neutralized $H_2C_2O_4$ particles remained effective.

Within two hours of uptake, the solid particles of $ox0.5_{70\%}$ experienced a morphological change and transformed into crystals (Fig. S2). Concurrently, the DMAH/NH$_4$ ratios dropped from 0.34 to <0.1 (Table 2, $t_{equil}$), showing that NH$_3$ can displace DMAH$^+$ from neutralized oxalate particles, which may trigger the transformation to a crystalline morphology. Final particle morphologies show a centered particle with a monoclinic or orthorhombic crystal structure, a shape typical of ammonium oxalate (Blake and Clegg, 2009), and small residuals of the satellite particles (Fig. S2).

### 3.4 Stability of reaction products and re-volatilization of NH$_3$ and DMA

As shown in Sect.1, single gas DMA uptake into $H_2SO_4$ equilibrated with incompletely neutralized droplets at both 10% and 50% RH with $X_{equil}$ between 1.5–1.9 (Table 1). Temporal profiles of the DMA uptake of 1ppm at 10 % RH show that DMAH/SO$_4$ ratios peaked after 4-5 hours (Fig. 1), reaching $X_{max\ neutral}$ of 1.7±0.1 before DMA partially re-volatilized and particles equilibrated with a more acidic composition of $X_{equil}$ = 1.5±0.1 (Table 1). When placing equilibrated samples into amine-free atmosphere, the neutralization ratio decreased further to $X_{N2}$ = 1.2±0.2 at 10% RH and $X_{N2}$ = 1.7±0.2 at 50% RH, as a result of DMA evaporation from these particles (Table 1). Similar degassing of methyl and ethyl-amines from synthesized salts has been reported in earlier studies (Chan and Chan, 2013; Chu et al., 2015; Lavi et al., 2015).

To examine the influence of NH$_4^+$ on the volatilization of DMA from particles and the formation of partially neutralized equilibrated droplets, we compared co-uptake experiments $0.5_{10\%}$ and $0.5_{50\%}$ to single gas uptake DMA$_{1ppm10\%}$, DMA$_{1ppm50\%}$. Figure 9 illustrates the maximum neutralization, equilibrium and composition of particles exposed to RH-conditioned N$_2$ atmosphere for the four mentioned experiments.

Particles in co-uptake experiments reached full neutralization (Fig. 9d and j). The total cation content in particles at 10% RH at the time of neutralization was comprised of > 80% NH$_4^+$ (Fig. 9d). When approaching equilibrium, DMAH$^+$ was displaced by NH$_3$ and the NH$_4^+$ content increased to 98% (Fig. 9e). Although the cation composition had changed, no decrease in $X$ beyond the margin of error was observed (Fig. 9d to e), even when particles were exposed to pure N$_2$ (Fig. 9f), due to the formation of a stable crystal. At 50% RH, DMA partially displaced NH$_4^+$ from neutralized particles, with an increase in DMAH/SO$_4$ from 0.7 at the point of maximum neutralization (Fig. 9j) to 1.3 at equilibrium (Fig. 9k). Accompanying the increase in DMAH$^+$ was a drop in the total neutralization ratios from $X$ = 2 to about $X$ = 1.8 (Fig. 9j to k), indicating that more NH$_3$ has degassed than was replaced by DMA. Hence while re-volatilization of DMA was responsible for the decrease in $X$ under DMA$_{1ppm50\%}$ conditions, degassing of NH$_3$ could be responsible for the decrease in neutralization ratios in co-uptake experiments. Once exposed to N$_2$, even more NH$_3$ evaporated from DMAH-NH$_4$ mixed particles (Fig. 9k to l), which further increased the acidity of these particles.

It should be noted, that the degassing of NH$_3$ was negligible for particles of the $0.1_{50\%}$ condition (Fig. 2c). At the point of maximum neutralization they contained large amounts of NH$_4^+$ and a DMAH/NH$_4$ ratio below 0.1, which seem to have prevented re-volatilization.

Under exposure to N$_2$, $X$ decreased in all particles that were in liquid phase state and contained large amounts of DMAH$^+$ (Fig. 9b, h, k), reflecting that the equilibrium compositions of these droplets were sensitive to changes in DMA and NH$_3$ gas concentrations.



### 4. Summary and Conclusions

The co-uptake of DMA and $NH_3$ into $H_2SO_4$ and $H_2C_2O_4$ particles was investigated at different RH and DMA/$NH_3$ gas ratios. The stoichiometric neutralization ratio and physical state of the particles were the two major factors influencing DMA and $NH_3$ uptake.

In the uptake into fresh, very acidic $H_2SO_4$ droplets at 10% and 50% RH, both DMA and $NH_3$ partitioned effectively, leading to a DMAH/$NH_4$ ratio comparable to the DMA/$NH_3$ gas ratio. Subsequently, the DMAH/$NH_4$ ratio decreased as $NH_3$ uptake was faster. The uptake of DMA and that of $NH_3$ were independent of each other because of the availability of abundant acids, as long as the particles did not reach neutralization nor undergo phase change. This result may explain why the highest particulate $R_3NH^+$ mass concentrations are detected in acidic aerosols with low $NH_4/SO_4$ (Youn et al., 2015).

In fully neutralized droplets at 50% RH, the limited availability of $H_3O^+$ ions for acid-base reactions led to a partial displacement of $NH_4^+$ by the stronger base DMA. This process yielded equilibrium particle compositions enriched in $DMAH^+$ by up to four times the gas phase ratio. It also potentially explains DMA partitioning into the neutralized condensed phase despite excess $NH_3$ (Sorooshian et al., 2008; Lloyd et al., 2009; Rehbein et al., 2011; VandenBoer et al., 2011).

At 10% RH, the phase changed from liquid to solid during uptake. This instantly inhibited further DMA uptake, while $NH_3$ uptake continued. Once the particles were fully neutralized, $NH_3$ displaced $DMAH^+$ from crystal structures and finally formed $(NH_4)_2SO_4$ with little residual $NH_4^+$, regardless of the DMA gas concentrations in the surrounding. $H_2C_2O_4$ particles at 70% RH were initially liquid, but transformed into non-crystalline solids after absorbing small amounts of DMA and $NH_3$. Subsequent uptake was similar to that of liquid sulfate particles. The formation of solid, partially neutralized $H_2C_2O_4$ particles did not hinder DMA uptake. Fully neutralized oxalate particles then crystallized upon displacement of $DMAH^+$ by $NH_3$, similar to the displacement of $DMAH^+$ from crystalline sulfate particles. Anhydrous $H_2C_2O_4$ at 10% RH was rather inert and took up small amounts of DMA and $NH_3$, presumably by adsorption only. In solid neutralized particles, $NH_3$ uptake is sterically and thermodynamically favored to form $(NH_4)_2SO_4$ or ammonium oxalate crystals by displacing $DMAH^+$. Hence, once ambient aerosols are in solid state, they are unlikely to take up $R_3N$, even when $R_3N$ and $NH_3$ gas concentrations are of the same magnitude.

In the absence of $NH_4^+$, DMA partially evaporated from $DMAH^+$–rich sulfate droplets. However, in the more common scenarios where $NH_4^+$ is present, DMA can displace $NH_4^+$ from neutral droplets, and cause additional $NH_3$ to evaporate and form non-neutralized particles. Hence, the presence of $DMAH^+$ can prevent aqueous sulfate particles from full neutralization. In our experiments $NH_3$ re-volatilization required DMAH/$NH_4$ ratio of about 0.5, which is at the upper end of DMAH/$NH_4$ measured in atmospheric particles (Sorooshian et al., 2008; VandenBoer et al., 2011; Youn et al., 2015).

The DMA and $NH_3$ gas concentrations and sulfate neutralization ratios used in the present study are high and are only likely in the vicinity of emission sources or in emission plumes (Sorooshian et al., 2008; Ge et al., 2011). Under such conditions, particle neutralization ratios are likely to be high (Sorooshian et al., 2008), and $NH_3$ and amines compete for particulate $H_3O^+$ ions, where DMA can displace $NH_4^+$ in liquid and $NH_3$ can displace $DMAH^+$ in solid particles. Although laboratory experiments have shown that in the absence of $NH_3$, DMA could partially displace $NH_4^+$ from solid ammonium salts including $(NH_4)_2SO_4$ and $NH_4NO_3$ (Lloyd et al., 2009; Qiu and Zhang, 2012; Chan and Chan, 2012, 2013), this scenario is unlikely under atmospheric conditions with abundant $NH_3$.

In this study, we used DMA as a proxy for atmospherically relevant $R_3N$. As different $R_3N$ and their sulfate and oxalate salts possess different hygroscopic and crystallization properties (Qiu and Zhang, 2012; Clegg et al., 2013; Chan and Chan, 2013;




Chu et al., 2015; Sauerwein et al., 2015) which can all influence uptake behavior, the findings here obtained may not be generalizable to all short-chain aliphatic amine compounds. It should also be mentioned that particle size may affect the particle composition as a result of the uptake, as well as the crystallization of the particle. The observed results may be most relevant to aerosols larger than 1 µm, which were found in the atmosphere to have the highest $R_3NH/NH_4$ ratio (VandenBoer et al., 2011; Youn et al., 2015). To improve our understanding of the mechanisms governing the simultaneous exchange of $NH_3$ and $R_3N$ between the gas and particle phases, particle size dependence should be investigated in the future.

### Acknowledgements

This work was supported by the Research Grants Council of the Hong Kong Special Administrative Region, China (GRF grant no. 600112 and 16300214).

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





**Table 1: Experimental conditions including relative humidity (RH), gas mixing ratios, the maximum neutralization ratios ($X_{max\ neutral}$) and neutralization ratios of equilibrated particles ($X_{equil}$), and the neutralization ratios of equilibrated particles exposed to $N_2$ for about 5 hours ($X_{N2}$).**

| Experiment | Particle composition | RH | [DMA] | [NH₃] | Neutralization ratio[a] | | |
|---|---|---|---|---|---|---|---|
| | | % | (ppm) | | $X_{max\ neutral}$[b] | $X_{equil}$ | $X_{N2}$ |
| $NH_{3,1.9ppm,10\%}$ | $H_2SO_{4\ (aq)}$ | 10 | | 1.94 ± 0.13 | | | |
| $NH_{3,1.9ppm,50\%}$ | $H_2SO_{4\ (aq)}$ | 50 | | 1.93 ± 0.12 | 2.0 ± 0.1 | 2.0 ± 0.1 | |
| $DMA_{0.15ppm,10\%}$ | $H_2SO_{4\ (aq)}$ | 10 | 0.16 ± 0.02 | | 1.5 ± 0.1 | 1.5 ± 0.1 | 1.2 ± 0.2 |
| $DMA_{0.15ppm,50\%}$ | $H_2SO_{4\ (aq)}$ | 50 | 0.16 ± 0.02 | | 1.7 ± 0.1 | 1.7 ± 0.1 | 1.7 ± 0.2 |
| $DMA_{1ppm,10\%}$ | $H_2SO_{4\ (aq)}$ | 10 | 1.00 ± 0.11 | | 1.7 ± 0.1 | 1.5 ± 0.1 | 1.2 ± 0.1 |
| $DMA_{1ppm,50\%}$ | $H_2SO_{4\ (aq)}$ | 50 | 1.00 ± 0.11 | | 1.9 ± 0.1 | 1.9 ± 0.1 | 1.7 ± 0.1 |
| $0.1_{10\%}$ | $H_2SO_{4\ (aq)}$ | 10 | 0.14 ± 0.01 | 1.89 ± 0.04 | 2.0 ± 0.1 | 2.0 ± 0.2 | 2.0 ± 0.4 |
| $0.1_{50\%}$ | $H_2SO_{4\ (aq)}$ | 50 | 0.14 ± 0.01 | 1.89 ± 0.04 | 2.0 ± 0.1 | 2.0 ± 0.2 | 1.9 ± 0.1 |
| $0.5_{10\%}$ | $H_2SO_{4\ (aq)}$ | 10 | 0.89 ± 0.03 | 1.93 ± 0.13 | 2.0 ± 0.1 | 1.9 ± 0.2 | 1.9 ± 0.2 |
| $0.5_{50\%}$ | $H_2SO_{4\ (aq)}$ | 50 | 0.89 ± 0.04 | 1.83 ± 0.04 | 2.0 ± 0.1 | 1.8 ± 0.1 | 1.7 ± 0.2 |
| $ox0.5_{10\%}$ | $H_2C_2O_{4\ (s)}$ | 10 | 0.89 ± 0.04 | 1.82 ± 0.03 | 0.3 ± 0.1 | 0.3 ± 0.1 | |
| $ox0.5_{70\%}$ | $H_2C_2O_{4\ (aq)}$ | 70 | 0.96 ± 0.00 | 1.86 ± 0.01 | 1.9 ± 0.2 | 1.9 ± 0.2 | |

5    [a] The neutralization ratio is defined as the number of moles of alkaline species over moles of acidic species, $X = (DMAH+NH_4)/SO_4$ or $X = (DMAH+NH_4)/C_2O_4$. [b] For co-uptake experiments $X_{max\ neutral}$ describes the maximum neutralization ratio of particles before reaching equilibrium.





**Table 2: Comparison of DMA/NH₃ (gas) molar ratios at different RH and DMAH/NH₄ (particle) ratios at different times (t), as well as the displacement percentage (χ) for DMAH⁺ (10% RH) and NH₄⁺ (50% RH) are given.**

| Experiment | RH | DMA/ NH$_3$ | DMAH/NH$_4$ | | | | | | | Displacement percentage [e] | |
|---|---|---|---|---|---|---|---|---|---|---|---|
| | % | | $t_{initial}$ [a] | | $t_{pc}$ [b] | | $t_{neutral}$ [c] | | $t_{equil}$ [d] | | $\chi_{DMAH+}$ | $\chi_{NH4+}$ |
| $0.1_{10\%}$ | 10 | 0.07 ± 0.01 | 0.07 ± 0.01 | 0.05 ± 0.01 | 0.03 ± 0.00 | 0.02 ± 0.00 | 42% | |
| $0.1_{50\%}$ | 50 | 0.07 ± 0.01 | 0.08 ± 0.01 | | 0.07 ± 0.01 | 0.18 ± 0.02 | | 9% |
| $0.5_{10\%}$ | 10 | 0.46 ± 0.04 | 0.43 ± 0.04 | 0.27 ± 0.01 | 0.19 ± 0.01 | 0.02 ± 0.00 | 89% | |
| $0.5_{50\%}$ | 50 | 0.49 ± 0.02 | 0.34 ± 0.02 | | 0.51 ± 0.04 | 1.77 ± 0.13 | | 50% |
| $ox0.5_{10\%}$ | 10 | 0.49 ± 0.02 | 0.25 ± 0.03 | | 0.25 ± 0.03 | 0.17 ± 0.04 | | |
| $ox0.5_{10\%}$ | 70 | 0.52 ± 0.52 | 0.34 ± 0.02 | 0.34 ± 0.02 | 0.26 ± 0.01 | 0.09 ± 0.02 | | |

[a] $t_{initial}$ indicates the time of the first measurement (10-15 min); [b] $t_{pc}$ indicates the time when the majority of particles changed from liquid to

5    solid phase, [c] $t_{neutral}$ indicates the time when particles reached (maximum) neutralization, [d] $t_{equil}$ indicates the time when the particle composition reached equilibrium, [e] The displacement percentage $\chi_{DMAH+}$ denotes $n$DMAH⁺ per particle at time $t_{pc}$ over time $t_{equil}$; $\chi_{NH4+}$ denotes $n$NH₄⁺ per particle at time $t_{neutral}$ over $t_{equil}$;





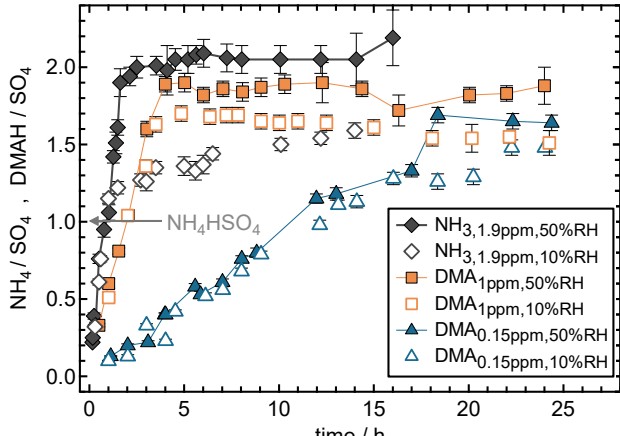

**Figure 1: Single gas uptake of NH₃ (1.9 ppm) and DMA (1 ppm and 0.15 ppm) into H₂SO₄, at 10% and 50% RH. NH₄/SO₄ and DMAH/SO₄ denote molar ratios of NH₃ or DMA species to total sulfate species in the particle.**





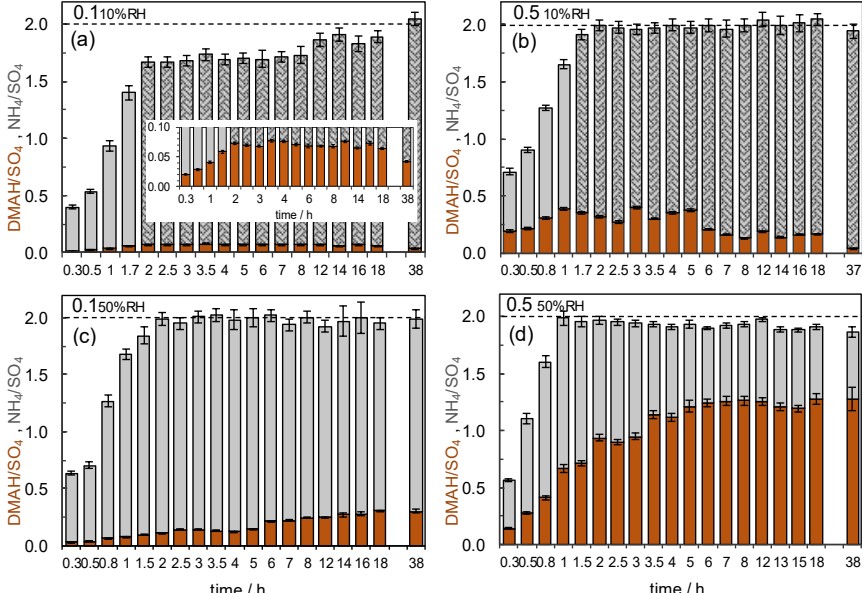

**Figure 2: DMA- NH₃ co-uptake of DMA and NH₃ into H₂SO₄ as a function of time: a) DMA/NH₃ = 0.07 at 10% RH (0.1₁₀%), b) DMA/NH₃ = 0.46 at 10% RH (0.5₁₀%), c) DMA/NH₃ = 0.07 at 50% RH (0.1₅₀%), and d) DMA/NH₃ = 0.49 at 50% RH (0.5₅₀%). A**

5   **value of two indicates complete neutralization of H₂SO₄. Contributions of DMAH⁺ (brown) and NH₄⁺ (grey) to the neutralization are shown as molar ratios (DMAH/SO₄ and NH₄/SO₄). Crosshatched NH₄⁺ bars indicate that the majority of particles underwent phase transition from liquid to solid. The inner graph in Fig. 2a is a magnified view of the DMAH⁺- fraction.**





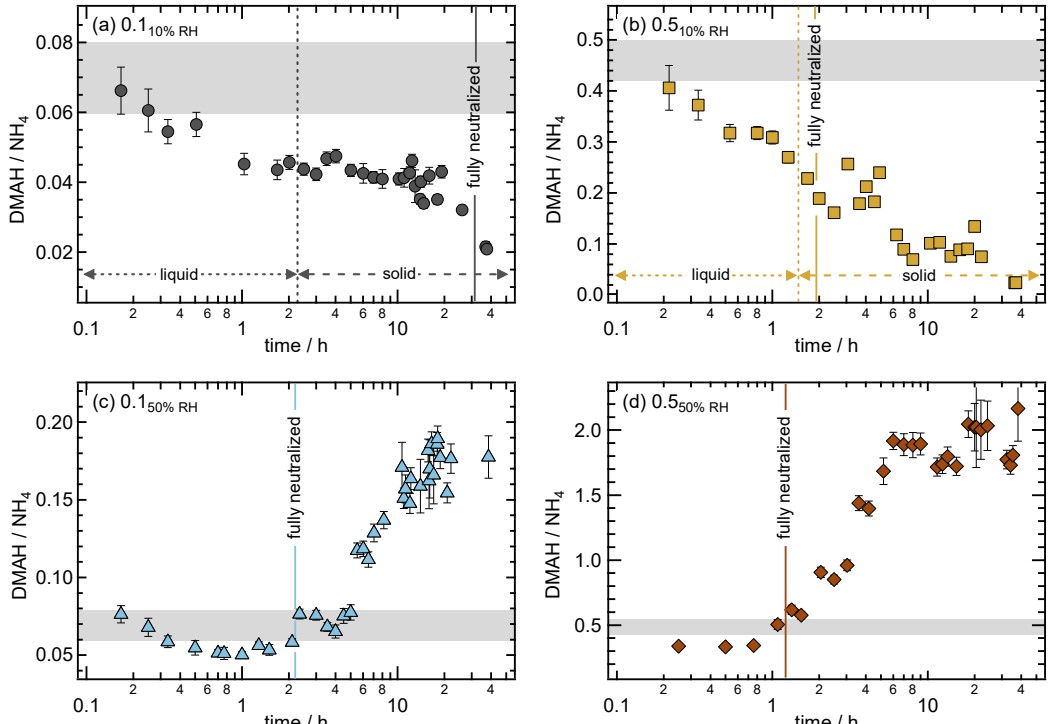

Figure 3: Particulate DMAH/NH$_4$ molar ratios as a function of time for DMA- NH$_3$ co-uptake at a) DMA/NH$_3$ = 0.07 at 10% RH (0.1$_{10\%}$), b) DMA/NH$_3$ = 0.46 at 10% RH (0.5$_{10\%}$), c) DMA/NH$_3$ = 0.07 at 50% RH (0.1$_{50\%}$), and d) DMA/NH$_3$ = 0.49 at 50% RH (0.5$_{50\%}$). DMA/NH$_3$ gas molar ratios (including uncertainties) are indicated by the grey bands. Dashed vertical lines indicate that the majority of particles underwent phase transition from liquid to solid. Solid vertical lines indicate that particles reached a neutralization ratio of two. Uncertainties from ion chromatography analysis are displayed on the vertical axis unless they are smaller than the symbols.



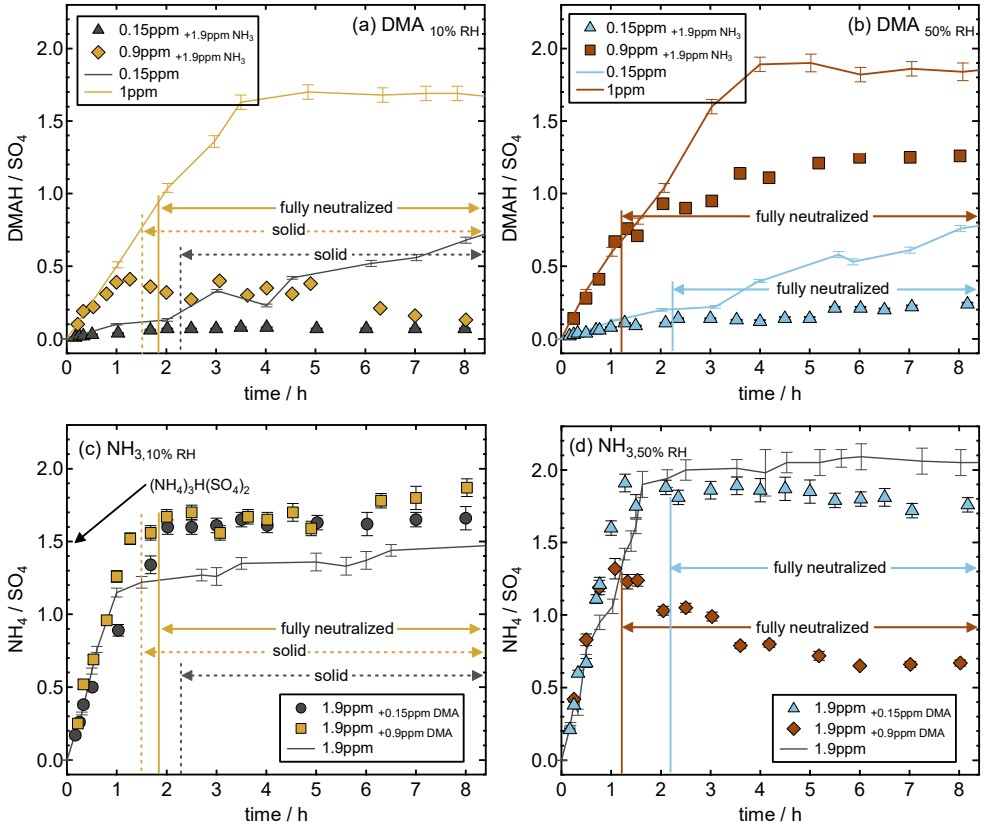

**Figure 4: Comparison of particulate DMAH/SO$_4$ (a, b) and NH$_4$/SO$_4$ (c, d) in single gas uptake (line) vs. DMA- NH$_3$ co-uptake (symbols) into H$_2$SO$_4$ at 10% RH (a,c) and 50% RH (b, d), in the first 8 hours of reaction. Dashed vertical lines indicate phase transition of the majority of particles. Solid vertical lines indicate that particles reached a neutralization ratio of two. Uncertainties from ion chromatography analysis are displayed on the vertical axis unless they are smaller than the symbols.**





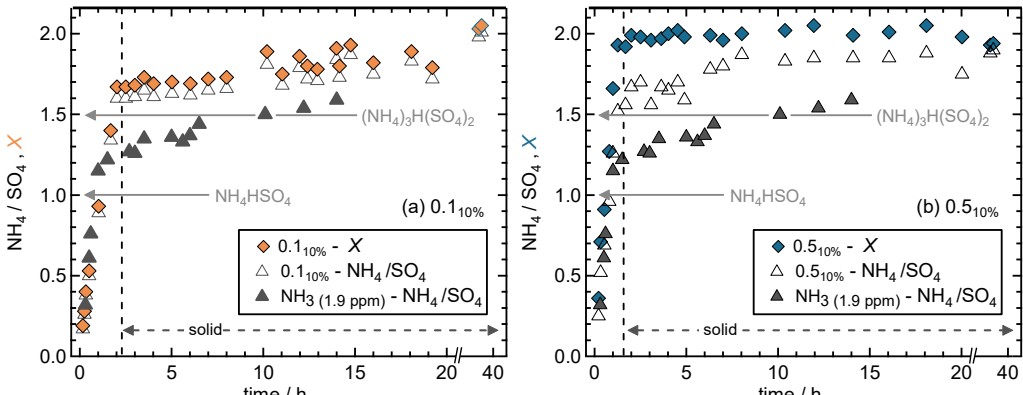

**Figure 5: Comparison of particulate NH₄/SO₄ in single gas uptake (filled triangles) vs. DMA- NH₃ co-uptake (open triangles) experiments at 10% RH into H₂SO₄: a) DMA/NH₃ = 0.07, and b) DMA/NH₃ = 0.46. Diamonds denote the net uptake ($X$) in co-uptake experiments. Error bars are omitted for clarity.**





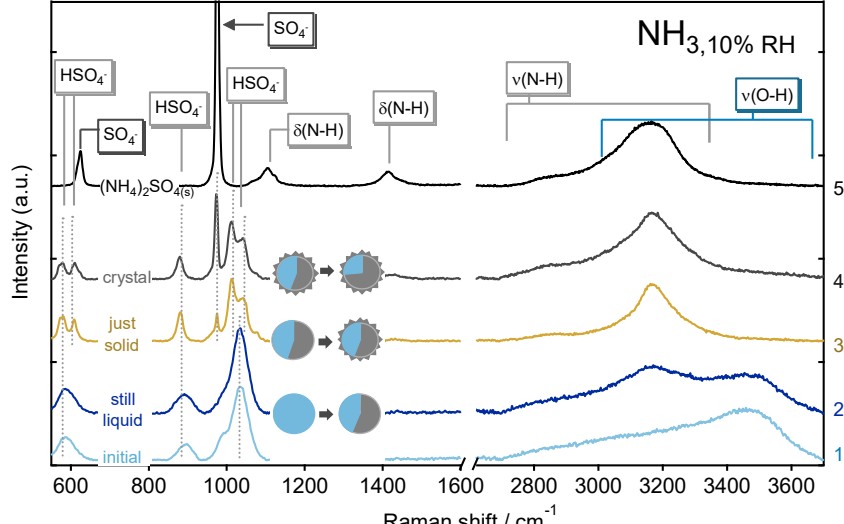

**Figure 6: Raman spectra of selected particles representing different phase states during uptake of $NH_3$ into $H_2SO_4$ at 10% RH. A spectrum of crystalline $(NH_4)_2SO_4$ is added for reference. Sketches indicate the cation composition range (as inferred from IC measurements) of particles when the spectra were recorded; grey denotes $NH_4^+$ and blue denotes $H_3O^+$ ($HSO_4^-$).**





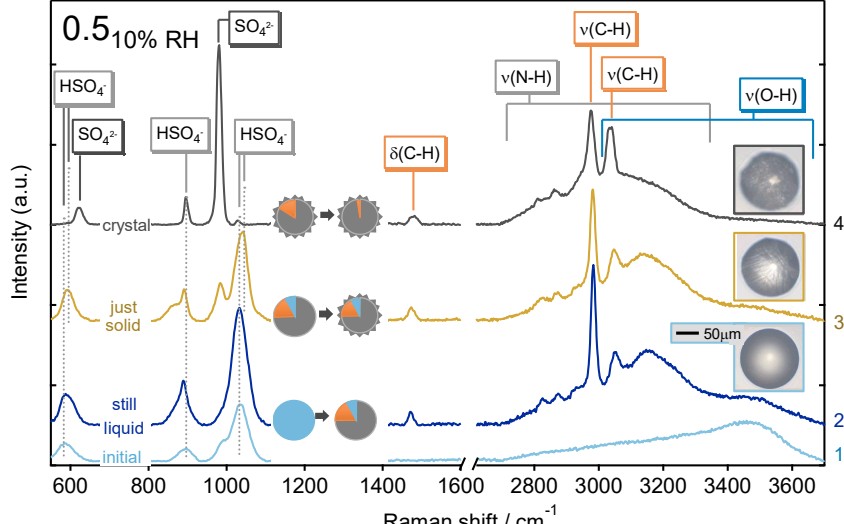

**Figure 7: Raman spectra and morphological changes of selected particles during DMA- NH$_3$ co-uptake into H$_2$SO$_4$ at 10% RH at a DMA/NH$_3$ molar ratio of 0.46 (0.5$_{10\%}$). Sketches indicate the cation composition range (as inferred from IC measurements) of particles when the spectra were recorded. Orange denotes DMAH$^+$, grey denotes NH$_4$$^+$, and blue denotes H$_3$O$^+$ (HSO$_4$$^-$).**





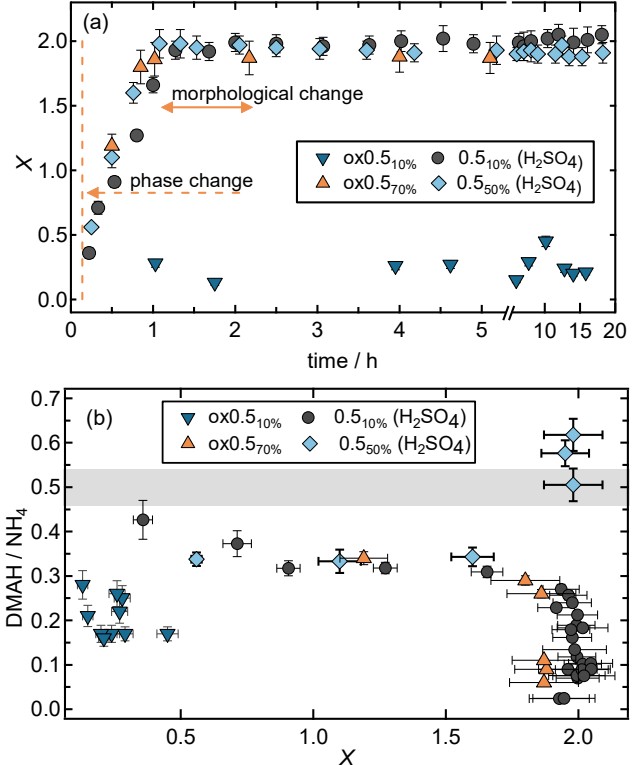

**Figure 8: DMA- NH₃ co-uptake at DMA/NH₃ gas molar ratios of 0.5 (given in Table 1) into sulfuric and oxalic acid particles at 10%, 50% and 70% RH. a) Neutralization ratios as a function of time. The vertical dotted line indicates that the majority of oxalic acid particles at 70% RH underwent phase transition from liquid to solid; the vertical solid line indicates the morphological transformation of oxalic acid particles from a non-crystalline to a crystalline morphology at 70% RH. b) DMAH/NH₄ ratios as a function of particle neutralization; the shaded area indicates the gas phase molar ratio of DMA/NH₃ (including uncertainties).**





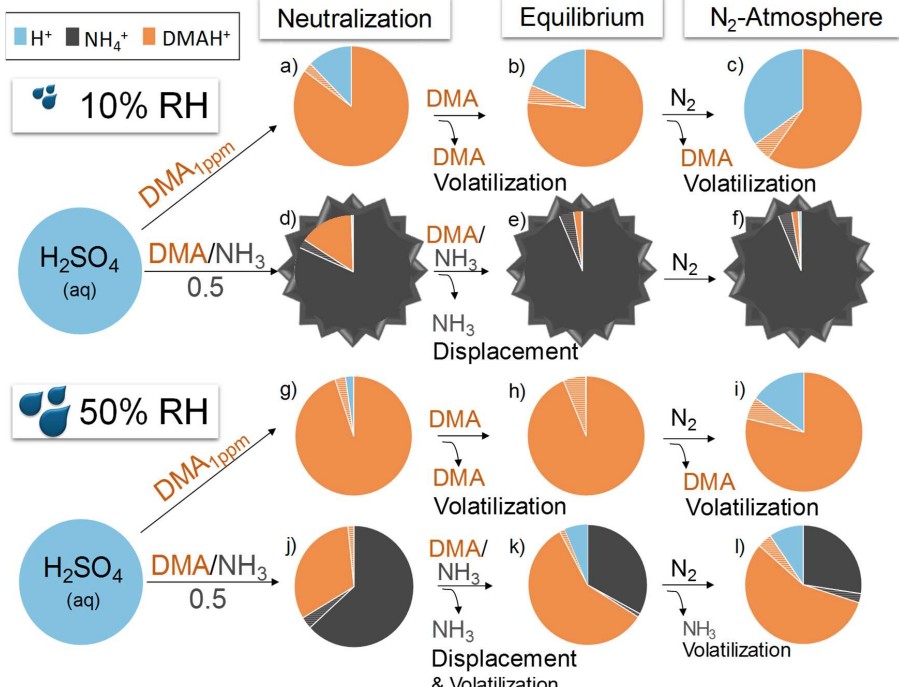

**Figure 9:** Particle neutralization ratios and the proportion of each cation at maximum neutralization (first column), equilibrium (second column), and after exposure of equilibrated particles to $N_2$ gas (third column). Color scheme: orange represents DMAH$^+$, grey represents NH$_4^+$, shaded areas indicate uncertainty of the respective species, and blue represents H$_3$O$^+$ (HSO$_4^-$). Stars around pie charts indicate that particles were crystalline solids. Concentrations of DMA and NH$_3$ are shown in Table 1.