# Peer review of "Heterogeneous uptake of ammonia and dimethylamine into sulfuric and oxalic acid particles"

_Atmospheric Chemistry and Physics, 2016_

## Referee Comment (RC1) · Anonymous Referee #1 · 12 Dec 2016

The manuscript describes time-dependent measurements of the competitive partitioning of ammonia and dimethyl amine to droplets/particles containing sulfuric acid or oxalic acid deposited on a substrate. Although many previous measurements have studied the partitioning of one of these gas phase species, this paper provides insight into the relative abundances of the species in the gas and condensed phase at different gas phase concentrations and relative humidities when acting in competition. The manuscript is helpful in providing a qualitative exploration of the problem although I believe some of the quantitative aspects need to be interpreted with some caution due to experimental uncertainties, particles sizes, etc. The inherently qualitative interpretation of the measurements does limit the more general application of the work

(and is sometimes quite hard to follow). However, the insights provided are sufficiently new that the manuscript merits publication - I do suggest the authors respond to the following suggestions before the paper can be accepted.

Page 5, line 40: The authors suggest that re-volatilization occurs at long times with the neutralization ratio decreasing from 1.7 to 1.5. Looking at the data in Figure 1, this seems to me to be over-interpreting the data. Firstly the "noise" in the calculated ratio seems to be larger than the error bars included with the points (e.g. see the fluctuations in the points around 17 hours for the DMA 0.15 ppm measurement at 50 % RH) – I suggest the authors consider again the uncertainties in their measurements. Secondly, how do the authors know that any such small change is not just due to a drift in environmental conditions (e.g. RH or temperature)? Infact, there is very little discussion of experimental errors in the analyses and the magnitude of uncertainties is quite key in verifying the veracity of trends identified in the data. I recommend a fuller discussion of experimental reproducibility and uncertainties.

Pages 6-7: I find the discussion of the change in neutralization and DMAH/NH4 ratios along with the surface acidity quite confusing. Given the large size of the droplets and the long timescales for the measurements, I can't see that different surface kinetic parameters (uptake coefficients) for the amine and ammonia can govern the change in these ratios. Indeed, I don't think this is what the authors are saying – if they are, then they need to consider competing rates of gas diffusion limited uptake, diffusional mixing within these liquid droplets, sensitivity of such large droplets to surface kinetics etc. Whichever is correct, the discussions is very confusing. This need for clarity is even more true when the authors then state that Figure 4 confirms that the uptake of DMA and NH3 proceed independently.

Page 12, lines 30-36: The authors consider the relevance of the high gas phase concentrations studied here when compared to atmospheric concentrations. It seems equally important to consider the importance of the particle size range studied. How do the authors expect their results to impact on our understanding of the much smaller

particles that must be considered in the atmosphere? There is a brief comment on this at the very end of the summary section.

Minor changes/corrections:

Page 3, line 20: The method for checking the gas phase ratios is not clearly described. The sentence describing this procedure needs to be expanded on: "To ensure accuracy of the gas ratio, measures were taken including conditioning the setup for a prolonged period, separating RH conditioning cells and reaction cells."

Page 6, line 6: Presumably the authors are referring to Sections 3.3 and 3.4 here?

Page 9, line 21: The error bar on this line is 0.00 – this is presumably not correct.

---

## Referee Comment (RC2) · Anonymous Referee #2 · 19 Jan 2017

The manuscript by Sauerwein and Chan describes results of laboratory experiments to investigate the competitive uptake of ammonia and amines in acidic particles. Experiments were conducted using dimethylamine as the representative alkyl amine and sulfuric or oxalic acid particles, at varying molar ratios of gas-phase amine/ammonia and varying relative humidities. Particulate aminium and ammonium molar ratios were measured by extraction followed by ion chromatography. The studies demonstrated uptake was influenced by the extent of neutralization and by the phase state of the particles. Some of the strengths of this work are that the experimental design considered: 1) co-uptake of ammonia and dimethylamine, 2) a range of relative humidities, and 3) a range of particle acidities. One weakness of this work was the use of large

particle sizes, though the reasons/limitations are clearly acknowledged and the results will nonetheless be useful to the community. One of the interesting findings was the influence of phase state on co-uptake and the effect of DMA/DMAH in limiting crystallization. The methodology and results were presented clearly, leaving no technical comments to be addressed. It is recommended that the minor editorial comments provided be addressed prior to publication.

Editorial Comments p2, line 10: The phrase "highest in marine particles as well as urban and rural aerosols" is confusing. It is not clear whether the authors are highlighting the importance of alkylaminium ions in marine aerosols, or the 140-560 nm size range.

p2, line 28: Do the particles actually absorb more water than ammonium sulfate particles across different compositions and sizes? Or are there some limits (e.g., only below the deliquescence point of ammonium sulfate)?

p3, line 30: Are the first experimental parameters for the sulfuric acid particles? If so, may want to specify that.

p6, line 4: Add '2-' to SO4 (as appears later in Section 3.2).

p6, section 3.2: Check section numbers here and throughout.

Section 3.2: NH3 displaced DMAH. . .should this be NH4? And then in paragraph below NH4 displaced by DMA. . .should be DMAH? Recommended to check throughout. Also recommended to check notation such as NH4 vs. NH4+, which appears to be used inconsistently.

p8, line 40: "During the"

Table 1: The subscript on the N looks like a superscript.

Fig. 2: x-axis font in panel b appears larger than in other panels. Some other inconsistencies between panels-recommended to check closely.

---

## Author Comment (AC1) · 23 Mar 2017

Meike Sauerwein[1] and Chak Keung Chan[1,2,3]

[1]Division of Environment, Hong Kong University of Science and Technology, Clear Water Bay, Kowloon, Hong Kong
[2]Department of Chemical and Biomolecular Engineering, Hong Kong University of Science and Technology, Clear Water
 Bay, Kowloon, Hong Kong
[3]School of Energy and Environment, City University of Hong Kong, Kowloon, Hong Kong

*Correspondence to*: Chak Keung Chan (Chak.K.Chan@cityu.edu.hk)

The authors would like to thank Anonymous Referee #1 for the comments on the manuscript. We respond to the specific comments made by the referee below and identify the changes we have done to the manuscript.

*Anonymous Referee #1: The manuscript describes time-dependent measurements of the competitive partitioning of ammonia and dimethyl amine to droplets/particles containing sulfuric acid or oxalic acid deposited on a substrate. Although many previous measurements have studied the partitioning of one of these gas phase species, this paper provides insight into the relative abundances of the species in the gas and condensed phase at different gas phase concentrations and relative humidities when acting in competition. The manuscript is helpful in providing a qualitative exploration of the problem although I believe some of the quantitative aspects need to be interpreted with some caution due to experimental uncertainties, particles sizes, etc. The inherently qualitative interpretation of the measurements does limit the more general application of the work (and is sometimes quite hard to follow). However, the insights provided are sufficiently new that the manuscript merits publication - I do suggest the authors respond to the following suggestions before the paper can be accepted.*

*Page 5, line 40: The authors suggest that re-volatilization occurs at long times with the neutralization ratio decreasing from 1.7 to 1.5. Looking at the data in Figure 1, this seems to me to be over-interpreting the data. Firstly the "noise" in the calculated ratio seems to be larger than the error bars included with the points (e.g. see the fluctuations in the points around 17 hours for the DMA 0.15 ppm measurement at 50 % RH) – I suggest the authors consider again the uncertainties in their measurements. Secondly, how do the authors know that any such small change is not just due to a drift in environmental conditions (e.g. RH or temperature)? Infact, there is very little discussion of experimental errors in the analyses and the magnitude of uncertainties is quite key in verifying the veracity of trends identified in the data. I recommend a fuller discussion of experimental reproducibility and uncertainties.*

**Response:** We thank the anonymous referee #1 for the valuable comment. Uncertainties as indicated by error bars in the figures originate from the analysis of particle composition with ion chromatography

analysis, while additional uncertainties that could explain fluctuations in particle phase concentration can only be described qualitatively. Those include:

1. Particle number - Variation in total particle count deposited on the hydrophobic film of 2000±100 droplets can cause a noticeable acceleration (if less particles were deposited) or delay (if more particles were deposited) in neutralization of the total sample including all particles.
2. Uncertainties resulting from variation in DMA/$NH_3$ gas ratios or gas concentration - Gas concentrations were estimated from regular gravimetric measurements of the permeation tubes to determine permeation rates. The small diffusion rates would lead to small changes between successive weight measurements and resulted in a few percent uncertainty. In addition, small variations in temperature (293.3±0.2 K) can cause small fluctuations in permeation rates (see additions made to the supplemental information). Though the uncertainty in gas concentrations was estimated and listed in Table 1, the effect of those uncertainties on the final particle phase concentration could not be quantified. Gas testing procedures and uncertainty estimations were added to the supplementary information.
3. Variations in temperature (296.3±1.0 K) and RH (10%±2% and 50%±3%) within the flow cells may have had small influences on uptake kinetics, and yet those uncertainties could not be further quantified.

In order to address the comments of Referee #1, in addition to the added information the supplement of the manuscript, we have added a qualitative description of the uncertainties into section 2.3, which reads as follows:

> "Uncertainties as shown in the figures were calculated based on errors resulting from IC measurements. Additional uncertainties of particle phase DMAH and $NH_4$ resulting from independent parameters such as the variations in total particle count, and uncertainties in generation and determination of gas concentrations also led to some fluctuations beyond the quantified errors."

The authors further agree with the referee's argumentation, that the decrease in neutralization ratio towards equilibrium lies within the margin of error if unquantifiable uncertainties are also considered. The paragraphs starting from line 38 - 41 on page 5, the sentence on line 11-13 on page 11, and the sentence in line 25-26 on page 12 were deleted. The subsequent sentence on page 11 line 14 was changed to

> "Under amine-free atmosphere, the neutralization ratio of these equilibrated particles decreased to $X_{N2} = 1.2\pm0.2$ at 10% RH and $X_{N2} = 1.7\pm0.2$ at 50% RH, as a result of DMA evaporation (Table 1).

On page 10, under consideration of uncertainties, we changed lines 23-25 to

> "DMAH/$NH_4$ ratios (Figure 8b) stayed roughly constant."

On page 11 line 39-41 was changed to

> "Under exposure to $N_2$, DMAH-$NH_4$ mixed particles that were originally in liquid phase state and contained large amounts of DMAH (Fig. 9 h, k) exhibited a decrease in $X$, reflecting that the equilibrium compositions of these droplets were sensitive to changes in DMA and $NH_3$ gas concentrations."

And the sentence on page 12 line 27 was moved to line 14 and altered as follows:

"When $NH_4$ is present in neutralized, DMAH–rich sulfate droplets, DMA from the surrounding gas phase can displace $NH_4$ from droplets, and prompt additional $NH_3$ to evaporate to form non-neutralized particles."

Despite the adjustments to more accurately describe the uncertainties of the measurements, we believe that the main conclusions of this work on the roles of particle phase and extent of neutralization in the co-uptake are not affected.

*Anonymous Referee #1: Pages 6-7: I find the discussion of the change in neutralization and DMAH/NH4 ratios along with the surface acidity quite confusing. Given the large size of the droplets and the long timescales for the measurements, I can't see that different surface kinetic parameters (uptake coefficients) for the amine and ammonia can govern the change in these ratios. Indeed, I don't think this is what the authors are saying – if they are, then they need to consider competing rates of gas diffusion limited uptake, diffusional mixing within these liquid droplets, sensitivity of such large droplets to surface kinetics etc. Whichever is correct, the discussions is very confusing. This need for clarity is even more true when the authors then state that Figure 4 confirms that the uptake of DMA and NH3 proceed independently.*

**Response:** We thank the anonymous referee #1 for pointing out those ambiguities. As mentioned at the end of section 3.2.1, the comparison of mixed gas with single gas uptake as shown in Figure 4 indicates that the initial $NH_3$ and DMA partitioning into acidic particles happens independently from each other. Thus, using uptake coefficients from earlier studies on single gas uptake to elucidate the mixed gas uptake process seems reasonable. Based on uptake coefficients reported in the literature which are larger for $NH_3$ than for DMA, one would expect the observed trend of the DMAH/$NH_4$ ratio to decrease as uptake proceeds since $NH_3$ uptake is faster than DMA. Yet it is unclear why the initial DMAH/$NH_4$ ratios (particle phase) are similar to DMA/$NH_3$ ratios (gas phase). Our discussion of surface acidity pertains to the very initial periods of uptakes.

The uptake coefficient describes the overall uptake process. This includes inter-related processes such as gas phase diffusion, immediate reaction of gas molecules colliding with the aerosol surface or adsorption and dissolution, followed by further liquid phase diffusion and proton transfer in the bulk particle.

Judging based on calculated mean thermal velocities $\omega_{NH3} = 607$ m s$^{-1}$ and $\omega_{DMA} = 373$ m s$^{-1}$ (296K) (based on diffusion coefficients of Massman, 1998 and Yaws 1995), $NH_3$ diffuses faster than DMA in gas phase. Yet, DMA possesses a higher gas phase and particle phase basicity than $NH_3$ (Brauman et al., 1971; Parrillo et al., 1993, Ge et al., 2011a) and a higher effective Henry's law constant than $NH_3$ for uptake into acidic solutions (Ge et al. 2011b) such as the sulfuric acid particles. Surface acidity was consequently considered to be one possible explanation why the stronger basicity of DMA may contribute to the equality of the DMAH/$NH_4$ in fresh particles and DMA/$NH_3$ ratios. This effect, however, diminishes as the particle gets gradually neutralized.

To add clarity to the discussion in section 3.2.1, we have added more description and rearranged paragraphs from lines 7-8 on page 7 as well as lines 34-40 from page 6 as follows:

"Starting from the second measured values, a clear decrease in DMAH/$NH_4$ ratios can be observed. Figure 4 compares the uptake of DMA and $NH_3$ in single and mixed gas experiments. The initial uptake trends of single gas and co-uptake do not deviate noticeably, indicating that DMA and $NH_3$ uptake took place independent of each other. Consequently, earlier reported

uptake coefficients from single gas uptake NH$_3$ and DMA into sulfuric acid may be used for co-uptake analysis."

And

"Wang et al. (2010) reported an uptake coefficient ($\gamma_{DMA}$) of about 0.03 for DMA uptake into concentrated H$_2$SO$_4$ of $\geq$ 62 wt% ($\leq$ 10% *RH*) at 283 K. The coefficient is noticeably smaller than that of close to unity reported for NH$_3$ uptake into H$_2$SO$_4$ of similar acidity (Swartz et al., 1999). In the current study, NH$_3$ uptake into fresh H$_2$SO$_4$ droplets was not overwhelmingly dominant. However, as the uptake continued, the DMAH/NH$_4$ ratios decreased by 30-40% for all experimental conditions within the first 1-2 hours (Fig. 3 a-d), which indicates a preferential uptake of NH$_3$. Since the gas concentrations of both NH$_3$ and DMA were constant, it is likely that the decreasing particle acidity and increasing neutralization ratio caused this change."

*Anonymous Referee #1: Page 12, lines 30-36: The authors consider the relevance of the high gas phase concentrations studied here when compared to atmospheric concentrations. It seems equally important to consider the importance of the particle size range studied. How do the authors expect their results to impact on our understanding of the much smaller particles that must be considered in the atmosphere? There is a brief comment on this at the very end of the summary section.*

When considering the application of the current results to submicron particles, two aspects might be of importance:

1.  The overall gas uptake would likely be accelerated. For submicron particles gas phase diffusion limitations would also have to be taken into consideration. Yet, we believe that the major findings in our paper i.e. the initially independent uptake of both gases into acidic particles as well as the roles of phase state and neutralization in DMAH or NH4 competition and displacement is not expected to change.
2.  Impact of particle size on crystallization behavior or inhibition of crystallization. DMAH, even if its concentration was low compared to NH$_4^+$ (DMAH/NH$_4$ = 0.05), was able to suppress NH$_4$HSO$_4$ precipitation. As it is generally easier to induce crystallization in larger particles, we assume that this effect should extend to particles of smaller sizes.

To stronger illustrate these aspects, the following sentence in the Summary and Conclusions section was modified to:

"It should also be mentioned that particle size may affect the gas uptake kinetics, as well as the crystallization behavior of the particle. When applying the results of this study to submicron particles, one may expect a considerably faster uptake and thus changes in DMAH/NH$_4$ ratios in acidic particles. Yet the overall trends of displacement reactions based on phase state and neutralization ratio as well as crystallization inhibition by DMAH are considered applicable to particles of smaller size. Nevertheless,[…]"

**Minor changes/corrections:**

*Anonymous Referee #1: Page 3, line 20: The method for checking the gas phase ratios is not clearly described. The sentence describing this procedure needs to be expanded on: "To ensure accuracy of the*

*gas ratio, measures were taken including conditioning the setup for a prolonged period, separating RH conditioning cells and reaction cells."*

**Response:** $NH_3$ and DMA are both very "sticky" gases (Robacker and Bartelt 1996, Hansen et al. 2013, Dawson et al. 2014), and are prone to adsorb to instrumental surfaces. Yet, the stickiness varies among different amines and ammonia (Namieśnik et al. 2003, Dawson et al. 2014). Measurements of the reestablishment of gas concentrations after the system underwent a cleaning process showed that the time to reach stable gas concentrations at the inlet of the flow cell took up to 4 hours for $NH_3$ and up to 8 hours for DMA. A much shorter recovery time was observed if only the gas supply was temporarily interrupted (e.g. for weighing permeation tubes, and/or changing the $N_2$ cylinder). In that case the time for re-establishment of stable concentrations took <1 hour for both gases. Since a drop in the concentration of one or both gases (e.g. due to wall losses) would result in a change of DMA/$NH_3$ ratio, we allowed 2 h equilibration time when the gas supply was interrupted and about 12 hours for reconditioning after the setup was cleaned. Since flow cells were likewise conditioned with DMA and $NH_3$ gas, we equilibrated the sulfuric acid or oxalic acid particles to the respective RH of 10% or 50% in separate clean cells.

In order to address the reviewers comment we have clarified the sentence as follows:

> "To ensure accuracy of the gas ratio, the system was conditioned for several hours to minimize wall losses either one or both gases prior to the uptake experiment (see supplemental information for detailed descriptions)."

We added more detail on the gas generation system, including the conditioning procedure, into the supplement of the paper.

***Anonymous Referee #1:*** *Page 6, line 6: Presumably the authors are referring to Sections 3.3 and 3.4 here?*

We thank the referee for this remark, the section numbering has been updated throughout the manuscript.

***Anonymous Referee #1:*** *Page 9, line 21: The error bar on this line is 0.00 – this is presumably not correct.*

We thank the referee for the comment. We increased the number of significant digits for this case in order to account for the uncertainties retrieved from IC. The sentence now reads as follows:

> "DMAH/$NH_4$ ratios at the time of neutralization ($t_{neutral}$) reached $0.032 \pm 0.001$ for the $0.1_{10\%}$ condition and $0.19 \pm 0.01$ for the $0.5_{10\%}$ condition (Table 2), showing a slight enrichment of $NH_4^+$ in the particle over the gas phase."

**References**

Brauman, J. I., Riveros, J. M., and Blair, L. K.: Gas-phase basicities of amines, J. Am. Chem. Soc., 93, 3914–3916, 1971.

Dawson, M.L., Varner, M.E., Perraud ,V., Ezell, M.J., Wilson J, Zelenyuk A, Gerber R.B., Finlayson-Pitts B.J.: Amine–Amine Exchange in Aminium–Methanesulfonate, Aerosols. J. Phys. Chem. C, 118 (50), 29431–29440, 2014.

Ge, X., Wexler, A. S., and Clegg, S. L.: Atmospheric amines – Part I. A review, Atmos. Environ., 45, 524–546, 2011a.

Ge, X., Wexler, A. S., and Clegg, S. L.: Atmospheric amines – Part II. Thermodynamic properties and gas/particle partitioning, Atmos. Environ., 45, 561–577, 2011b.

Hansen, M.J., Adamsen, A.P.S., Feilberg, A.: Recovery of odorants from an olfactometer measured by proton-transfer-reaction mass spectrometry. Sensors, 13(6), 7860–7871, 2013.

Massman, W.J.: A review of the molecular diffusivities of $H_2O$, $CO_2$, $CH_4$, CO, $O_3$, $SO_2$, $NH_3$, $N_2O$, NO, and $NO_2$ in air, $O_2$ and $N_2$ near STP, Atmos. Environ., 32, 1111-1127, 1998 .

Namieśnik J., Jastrzębska, A., Zygmunt, B.: Determination of volatile aliphatic amines in air by solid-phase microextraction coupled with gas chromatography with flame ionization detection. J. Phys. Chem. A, 1016(1), 1–9, 2003.

Parrillo, D. J., Gorte, R. J., and Farneth, W. E.: A calorimetric study of simple bases in H-ZSM-5: A comparison with gas-phase and solution-phase acidities, J. Am. Chem. Soc., 115, 12441–12445, 1993.

Robacker D.C., Bartelt R.J.: Solid-Phase Microextraction Analysis of Static-Air Emissions of Ammonia, Methylamine, and Putrescine from a Lure for the Mexican Fruit Fly (Anastrephaludens ), J. Agric. Food Chem., 44(11), 3554–3559, 1996.

Yaws C.L.: Handbook of Vapor Pressure: Inorganic Compounds and Elements, Volume 4, Gulf Professional Publishing, 1995.

---

## Author Comment (AC2) · 23 Mar 2017

Meike Sauerwein[1] and Chak Keung Chan[1,2,3]

[1]Division of Environment, Hong Kong University of Science and Technology, Clear Water Bay, Kowloon, Hong Kong
[2]Department of Chemical and Biomolecular Engineering, Hong Kong University of Science and Technology, Clear Water
 Bay, Kowloon, Hong Kong
[3]School of Energy and Environment, City University of Hong Kong, Kowloon, Hong Kong

*Correspondence to*: Chak Keung Chan (Chak.K.Chan@cityu.edu.hk)

The authors would like to thank Anonymous Referee #2 for the comments on the manuscript. We respond to the specific comments made by the referee below and identify the changes we have done to the manuscript.

*Anonymous Referee #2: p2, line 10: The phrase "highest in marine particles as well as urban and rural aerosols" is confusing. It is not clear whether the authors are highlighting the importance of alkylaminium ions in marine aerosols, or the 140-560 nm size range.*

**Response:** We thank the anonymous referee #2 for the comment. To improve clarity the sentence was changed to

> "In fact, mass loadings of alkylaminium ions ($R_3NH^+$) are the highest in particles with a diameter of 140-560 nm, both in urban and rural, as well as in marine aerosols (Müller et al., 2009; VandenBoer et al., 2011; Youn et al.,2015)."

*Anonymous Referee #2: p2, line 28: Do the particles actually absorb more water than ammonium sulfate particles across different compositions and sizes? Or are there some limits (e.g., only below the deliquescence point of ammonium sulfate)?*

**Response:** Recent studies on the hygroscopicity of methyl- and ethyl-aminium sulfate salts and their mixtures with sulfuric acid (Chu et al. 2015, Sauerwein et al. 2015, Rovelli et al. 2016) have shown that short-chain alkyl-aminium sulfates are significantly more hygroscopic than ammonium sulfate up to 95% RH, hence beyond the deliquescence point of ammonium sulfate. In all three studies, hygroscopicity of alkyl-aminium sulfates was shown to increase from primary to tertiary aminium sulfate. The difference in hygroscopicity decreases as extent of neutralization decreases to bisulfate composition, where aminium and ammonium bisulfate possess similar hygroscopic properties (Sauerwein et al. 2015).

*Anonymous Referee #2: p3, line 30: Are the first experimental parameters for the sulfuric acid particles? If so, may want to specify that.*

**Response:** We agree with anonymous referee #2. To no leave any ambiguity, we added the following for clarification

> "Combination of the DMA flow (0.15 or 0.9-1.0 ppm) and the $NH_3$ flow (1.8-1.9 ppm) resulted in $DMA/NH_3$ ratios of $0.07\pm0.01$ and $0.46\pm0.04$ at 10% RH, as well as $0.07\pm0.01$ and $0.49\pm0.02$ at 50% RH for experiments with $H_2SO_4$ (Table 2), and $0.49\pm0.02$ at 10% and $0.52\pm0.01$ at 70% RH for experiments with $H_2C_2O_4$."

Further changes were made as suggested by anonymous referee #2 as follows:

***Anonymous Referee #2:*** *p6, line 4: Add '2-' to SO4 (as appears later in Section 3.2).*

**Response:** Changes were made as follows:

> "Figure 2 depicts the temporal profiles of $DMAH/SO_4$, $NH_4/SO_4$ and *X* at the different gas ratios and RH. At 10% RH particles solidified during the experiment (Fig. 2a and b, indicated by crosshatched areas) and needed 2 to > 18 hours (for $0.5_{10\%}$ and $0.1_{10\%}$, respectively) to completely neutralize sulfate."

***Anonymous Referee #2:*** *p6, section 3.2: Check section numbers here and throughout.*

**Response:** We thank the referee for this comment, the section numbering has been updated throughout the manuscript.

***Anonymous Referee #2:*** *Section3.2: $NH_3$ displaced DMAH...should this be $NH_4$? And then in paragraph below NH4 displaced by DMA...should be DMAH? Recommended to check throughout.*

**Response:** We agree with anonymous referee #2 that the displacement of one alkaline species by another alkaline gas is inseparable connected with the gas absorption and protonation, and hence technically the argumentation, that $DMAH^+$ is displaced by $NH_4^+$ is correct. Yet, the authors like to follow the terminology that is commonly used in literature focused on similar kind of displacement reactions such as Qiu et al. 2011.

***Anonymous Referee #2:*** *Also recommended to check notation such as NH4 vs. NH4+, which appears to be used inconsistently.*

**Response:** Following the definition in section 2.3, we use DMAH to represent particle phase $nNH_2(CH_3)_2^+ + nNH(CH_3)_2$, $NH_4$ to represent $nNH_4^+ + nNH_3$, $SO_4$ to indicate total amount of sulfate species $= nH_2SO_4 + nHSO_4^- + nSO_4^{2-}$, and $C_2O_4$ to indicated total amount of oxalate species (*n* denotes the molar amounts of each compound). The manuscript was reviewed and expressions such as $NH_4^+$ and $NH_4$-species were consistently named as $NH_4$, the same was done for DMAH.

***Anonymous Referee #2:*** *p8, line 40: "During the"*

**Response:** Following the suggestion the sentence was rewritten as:

"During $NH_3$-DMA co-uptake the absorbed DMAH seems to have suppressed the precipitation of $NH_4HSO_4$ in particles with a composition of $1.1 < NH_4/SO_4 < 1.5$, so that the phase change started only at a $NH_4/SO_4$ around 1.5 (Fig. 5a and b)."

*Anonymous Referee #2:* *Table 1: The subscript on the N looks like a superscript.*

**Response:** the term $X_{N2}$ in Table 1 and the table caption have been updated to $X_{N_2}$.

*Anonymous Referee #2:* *Fig. 2: x-axis font in panel b appears larger than in other panels. Some other inconsistencies between panels-recommended to check closely*

**Response:** The figure has been updated as shown below

[Figure]

**References**

Chu, Y., Sauerwein, M., and Chan, C. K.: Hygroscopic and phase transition properties of alkyl aminium sulfates at low relative humidities, Phys. Chem. Chem. Phys., 17, 19789–19796, 2015.

Qiu, C., Wang, L., Lal, V., Khalizov, A. F., and Zhang, R.: Heterogeneous reactions of alkylamines with ammonium sulfate and ammonium bisulfate, Environ. Sci. Technol., 45, 4748–4755, 2011.

Rovelli, G., Miles, R. E. H., Reid, J. P., and Clegg, S. L.: Hygroscopic Properties of Aminium Sulphate Aerosols, Atmos. Chem. Phys. Discuss., in review, 2016.

Sauerwein, M., Clegg, S. L., and Chan, C. K.: Water Activities and Osmotic Coefficients of Aqueous Solutions of Five Alkylaminium Sulfates and Their Mixtures with $H_2SO_4$ at 25°C, Aerosol Sci. Tech., 49, 566–579, 2015.